# Towards Provable Log Density Policy Gradient

**Pulkit Katdare**
**University of Illinois at Urbana-Champaign**

**Anant A. Joshi**
**University of Illinois at Urbana-Champaign**

**Katherine Driggs-Campbell**
**University of Illinois at Urbana-Champaign**

**Reviewed on OpenReview:** `https://openreview.net/pdf?id=qIWazsRaTR`

## Abstract

Policy gradient methods are a vital ingredient behind the success of modern reinforcement learning. Modern policy gradient methods, although successful, introduce a residual error in gradient estimation. In this work, we argue that this residual term is significant and correcting for it could potentially improve sample-complexity of reinforcement learning methods. To that end, we propose log density gradient to estimate the policy gradient, which corrects for this residual error term. Log density gradient method computes policy gradient by utilising the state-action discounted distributional formulation. We first present the equations needed to exactly find the log density gradient for a tabular Markov Decision Processes (MDPs). For more complex environments, we propose a temporal difference (TD) method that approximates log density gradient by utilizing backward on-policy samples. Since backward sampling from a Markov chain is highly restrictive we also propose a min-max optimization that can approximate log density gradient using just on-policy samples. We also prove uniqueness, and convergence under linear function approximation, for this min-max optimization. Finally, we show that the sample complexity of our min-max optimization to be of the order of $m^{-1/2}$, where $m$ is the number of on-policy samples. We also demonstrate a proof-of-concept for our log density gradient method on gridworld environment, and observe that our method is able to improve upon the classical policy gradient method by a clear margin, thus indicating a promising novel direction to develop reinforcement learning algorithms that require fewer samples.

## 1 Introduction

Policy gradient (PG) methods are a vital ingredient behind the success of modern reinforcement learning (Silver et al., 2017; John Schulman et al., 2023; Haarnoja et al.; Kakade, 2001). The success of PG methods stems from their simplicity and compatibility with neural network-based function approximations (Sutton et al., 1999; Baxter & Bartlett, 2001). Although modern policy gradient methods like PPO and TRPO, have achieved excellent results in various on-policy tasks (Schulman et al., 2017; 2015), they require extensive hyper-parameter tuning. Additionally, it has been shown by Ilyas et al. (2020) that the estimation error between policy gradient estimated by the methods like PPO and the true policy gradient increases significantly during the training process. Classically in reinforcement learning, two types of problems are well known and well studied: discounted reward and average reward, see for example Sutton & Barto (2018). It is well known that for the discounted reward scenario, the Bellman operator is a contraction, and solving the Bellman equation iteratively is a straightforward numerical procedure. However, the same contraction property fails to hold for the average reward scenario. Therefore, classical policy gradient methods typically approximate gradient of the policy using Q-function estimated with discount factor strictly less than 1, which leads to a

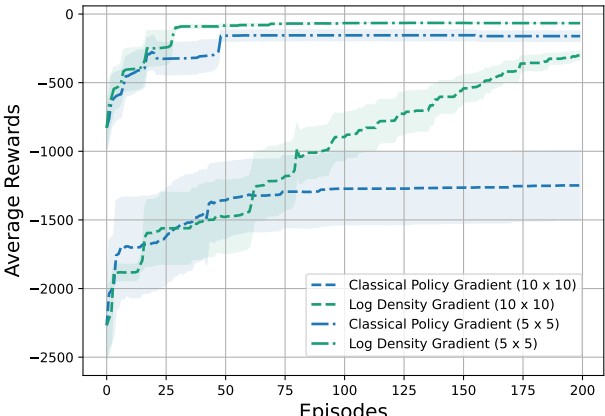

Figure 1: For the average reward scenario, performance of classical policy gradient (blue) algorithm as compared to log density gradient (green) algorithm over a $n \times n$ gridworld environment, for $n = 5, 10$. We observe that log density gradient algorithm consistently converges to better policy performance. Theoretical calculated solutions are used for implementation.

error in gradient estimation (Morimura et al., 2010). In this paper, we empirically demonstrate that this error in indeed significant, see for instance, Figure 1. We further propose a novel algorithm to estimate policy gradient that corrects for this residual error, which could potentially lead to sample efficient reinforcement learning, thus enabling their deployment over a wide variety of complex scenarios. We call our method, log density gradient. We show that log density gradient method can be used to estimate the policy gradient for all values of discounting factor – including the *average reward scenario.* Log density gradient method is based on the average state-action stationary distribution formulation of reinforcement learning, which allows for the estimation of policy gradient as a multiplication of the gradient of log density and the reward function (Nachum et al., 2019; Uehara et al.). This separation results in an improved correlation with the true policy gradient and requires fewer hyperparameters. We show that our method is consistent with the classical policy gradient theorem (Sutton, 1988) and also prove convergence properties and sample complexity.

Our main contributions are as follows. **1.** A novel method to *provably* calculate policy gradient by using the average state-action discounted formulation for all values of the discounting factor. We will show that policy gradient estimated in this manner for average reward scenario will correct for the residual error in policy gradient estimation, which is widely ignored in empirical implementations of policy gradients (as shown in Figure 1). **2.** A model-free Temporal Difference (TD) method for approximating policy gradient. We provide proof of contraction as well as convergence. However, there is a major drawback that it requires samples from the backward Markov chain (described in detail in the paper) which motivates the next contribution. **3.** A min-max optimization which yields the gradient of log density for all values of the discounting factor including the average reward scenario and a model free TD method to implement it, with proof of convergence. We also show that this min-max optimization has a closed form solution under linear function class assumptions, thus enabling their practical use with linear MDP problems (Zhang et al., 2022). We additionally show sample complexity of the order $O(m^{-1/2})$ for the projected version of the proposed TD method, where $m$ is the number of on-policy samples. Our method is competitive with the sample complexity of classical vanilla policy gradient methods (Yuan et al., 2022).

Section 2 starts with problem formulation and motivation behind this paper. Section 3, discusses prior work in policy gradient methods, temporal difference methods, and min-max problems in off-policy evaluation and compares our work with existing works to situate our paper in the literature. Our main contributions are discussed in detail starting from Section 5 which starts with rigorously defining log density gradient. Additionally we also propose a TD approach to estimate log density gradient under strict reversibility assumptions, and we describe the issue caused by this assumption. In section 6, to overcome this issue to

propose a min-max variant that allows us to estimate log density gradient algorithm using empirical samples. We finally demonstrate a proof-of-concept of our algorithm in Section 7 which shows that log density gradient can be potentially sample efficient as compared to classical policy gradient methods.

## 2 Background and Motivation

**Notation:** we let $(\cdot)^T$ denote matrix transpose, and let $e$ represent the vector of ones, the size of which would be clear from context.

We define Markov Decision Process (MDP) as a 6-tuple of $(\mathcal{S}, \mathcal{A}, \mathcal{P}, r, \gamma, d_0)$. Here, $\mathcal{S}$ is a finite state space of the MDP, $\mathcal{A}$ is a finite action space, $\mathcal{P}$ is the transition probability matrix, $r$ is the reward function and $d_0$ is the initial distribution. The reinforcement learning problems is optimise for a policy $\pi : \mathcal{S} \to \Delta(\mathcal{A})$ that maximizes $J_\gamma(\pi)$, defined as

$$J_\gamma(\pi) := (1 - \gamma)\mathbb{E}[\sum_{t=0}^{\infty} \gamma^t r(s_t, a_t)|s_0 \sim d_0, a_t \sim \pi(\cdot|s_t), s_{t+1} \sim \mathcal{P}(\cdot|s_t, a_t)], \text{ for } \gamma \in [0, 1)$$

$$J_1(\pi) := \lim_{\mathcal{T} \to \infty} \mathbb{E}[\frac{1}{\mathcal{T}} \sum_{t=0}^{\mathcal{T}} \gamma^t r(s_t, a_t), s_0 \sim d_0, a_t \sim \pi(\cdot|s_t), s_{t+1} \sim \mathcal{P}(\cdot|s_t, a_t)], \text{ for } \gamma = 1,$$

where $\gamma \in [0, 1]$ is the discounting factor which accounts for the impact of future rewards in present decision making. When $\gamma = 1$, $J_1(\pi)$ the scenario is called the average reward formulation. Most practical problems in reinforcement learning typically aim to solve for an optimal policy $\pi^* = \arg\max_\pi J_1(\pi)$ (See Figure 1 Haarnoja et al.). Modern reinforcement learning methods aim to parameterise policy with a set of parameters $\theta \in \mathbb{R}^n$, where $n$ is the dimensions of the parameter space. We refer to such paremterisation as $\pi_\theta$. This kind of parameterisation enables us search for optimal set of parameters $\theta^*$ instead of a search over $\mathcal{S} \times \mathcal{A}$ which in practice could be very large. We define

$$\theta^* := \arg\max_{\theta \in \mathbb{R}^n} J_1(\pi_\theta).$$

The Q-function $Q_\gamma^{\pi_\theta}$ is commonly used function used to describe the performance of an RL agent. Q-function calculates the long term (discounted) rewards accumulated by an agent following a fixed policy $\pi_\theta$ while starting from a state $s \in \mathcal{S}$ and taking an action $a \in \mathcal{A}$

$$Q_\gamma^{\pi_\theta}(s, a) := \mathbb{E}[\sum_{t=0}^{\infty} \gamma^t r(s_t, a_t)|s_0 = s, a_0 = a, a_t \sim \pi_\theta(\cdot|s_t), s_{t+1} \sim \mathcal{P}(\cdot|s_t, a_t)] \tag{1a}$$

$$= r(s, a) + \gamma\mathbb{E}_{s' \sim \mathcal{P}(\cdot|s,a), a' \sim \pi_\theta(\cdot|s')}[Q_\gamma^{\pi_\theta}(s', a')] \tag{1b}$$

Where, equation 1b is called the Bellman Equation. Bellman equation is popularly used to estimate the Q-function using just empirical data collected on the MDP. Q-function approximation methods typically use $\gamma < 1$ for stable estimation of the Q-function. We also similarly define value function $V_\gamma^{\pi_\theta}(s) = \mathbb{E}_{a \sim \pi_\theta(\cdot|s)}[Q_\gamma^{\pi_\theta}(s, a)]$.

Modern RL algorithms generally solve for $\theta^*$ by estimating the gradient of policy performance $J_\gamma(\pi_\theta)$ with respect to policy parameters $\theta$. This is also commonly referred to as the policy gradient theorem $\nabla_\theta J_1(\pi_\theta)$ (Sutton et al., 1999) which says

$$\nabla_\theta J_\gamma(\pi_\theta) = \mathbb{E}_{(s,a) \sim d_\gamma^{\pi_\theta}}[Q_\gamma^{\pi_\theta}(s, a) \cdot \nabla_\theta \log \pi_\theta(a|s)], \quad \gamma \in [0, 1]. \tag{2}$$

Here, $d_\gamma^{\pi_\theta}$ is the average state-action discounted stationary distribution, which is defined as the cumulative sum of discounted state-action occupancy across the time horizon.

$$d_\gamma^{\pi_\theta}(s, a) := (1 - \gamma) \sum_{t=0}^{\infty} \gamma^t \mathbb{P}(s_t = s, a_t = a|s_0 \sim d_0, a_t \sim \pi_\theta(s_t), s_{t+1} \sim \mathcal{P}(\cdot|s_t, a_t), \gamma < 1) \tag{3a}$$

$$d_1^{\pi_\theta}(s, a) := \lim_{T \to \infty} \frac{1}{T} \sum_{t=0}^{T} \mathbb{P}(s_t = s, a_t = a|s_0 \sim d_0, a_t \sim \pi_\theta(s_t), s_{t+1} \sim \mathcal{P}(\cdot|s_t, a_t), \gamma = 1). \tag{3b}$$

In this paper we make a standard assumption that the Markov chain induced by policy $\pi_\theta$ is ergodic. In particular, this implies that $d_\gamma^{\pi_\theta}(s, a) > 0$ for all state action pairs $(s, a)$ (Puterman, 2014). In scenarios where we are trying to optimize for $J_1(\pi)$, estimating the policy gradient becomes difficult. This is because the Bellman equation cannot be used to estimate Q-function for $\gamma = 1$. As a compromise policy gradient for average reward scenarios are instead approximated by calculating the Q-function for a discounting factor $\gamma < 1$, but close to 1, and using that estimate in the policy gradient equation 2

$$\hat{\nabla}_\theta J_1(\pi_\theta) = \mathbb{E}_{(s,a)\sim d_1^{\pi_\theta}}[Q_1^{\pi_\theta}(s, a) \cdot \nabla_\theta \log \pi_\theta(a|s)], \tag{4a}$$

$$\approx \mathbb{E}_{(s,a)\sim d_1^{\pi_\theta}}[Q_\gamma^{\pi_\theta}(s, a) \cdot \nabla_\theta \log \pi_\theta(a|s)], \quad \gamma < 1. \tag{4b}$$

In this paper we argue that the policy gradient calculated in this manner induces a significant residual error, which keeps on compounding as the reinforcement learning training proceeds even leading to a sub optimal solution. The following equation, derived in Proposition 2 characterizes that error,

$$\nabla_\theta J_1(\pi_\theta) = \mathbb{E}_{(s,a)\sim d_1^{\pi_\theta}}[Q_\gamma^{\pi_\theta}(s, a) \cdot \nabla_\theta \log \pi_\theta(a|s)] + \underbrace{(1-\gamma)\mathbb{E}_{(s,a)\sim d_1^{\pi_\theta}}[\nabla_\theta \log d_1^{\pi_\theta}(s) \cdot V_\gamma^{\pi_\theta}(s)]}_{\textbf{Residual Error}} \tag{5}$$

In this paper, we prove that this residual error is significant. What more, we also propose another method to exactly obtain the policy gradient, for all values of the discounting factor including $\gamma = 1$ which we call as the log density gradient. Our estimation of log density gradient utilises average state-action discounted distributional formulation of a reinforcement learning problem which re-states $J_\gamma(\pi)$ as expectation under $d_\gamma^{\pi_\theta}$ (Nachum et al., 2019; Uehara et al.) as

$$J_\gamma(\pi) = \mathbb{E}_{(s,a)\sim d_\gamma^{\pi_\theta}}[r(s, a)].$$

Under this formulation, policy gradient can similarly be obtained using log derivative trick as follows,

$$\nabla_\theta J_\gamma(\pi) = \mathbb{E}_{(s,a)\sim d_\gamma^{\pi_\theta}}[\nabla_\theta \log d_\gamma^{\pi_\theta}(s, a) \cdot r(s, a)]. \tag{6}$$

We refer to $\nabla_\theta \log d_\gamma^{\pi_\theta}$ as the log density gradient. A key advantage of log density gradient is that it would allow us to approximate policy gradient for average reward scenarios in a provable manner. In this work, we show that log density gradient can be approximated even under average reward scenarios ($\gamma = 1$).

## 3 Survey of Related Work and Comparison

In this section we will discuss existing studies in policy gradient methods including the framework of log density gradient first introduced by Morimura et al. (2010). We also briefly discuss density ratio learning methods which have been very popular in off-policy evaluation. A short discussion on Temporal Difference (TD) learning methods may be found in Appendix 9.1.

### 3.1 Policy Gradient Methods

**Literature survey:** Policy gradient methods are a widely studied topic in reinforcement learning. One of the earliest works in this area proposed a closed-form solution for evaluating policy gradients called the policy gradient theorem (Sutton et al., 1999). Initially implementations for policy gradient methods used episodic estimates to update policy parameters (Williams, 1992) and GPOMDP (Baxter & Bartlett, 2001). Unfortunately this way of implementing policy gradient suffered from high variance, thus inhibiting scalability to large problem spaces (Schulman et al., 2016). To address this problem, actor-critic methods approximate the Q-function or advantage function using an additional neural network, which are then used to update the policy (Mnih et al., 2016; Schulman et al., 2016). Furthermore policy gradient methods are also designed to be compatible with deterministic policies (Lillicrap et al., 2016; Silver et al., 2014). Recently Trust region methods, such as Trust Region Policy Optimization (TRPO) Schulman et al. (2015) and Proximal Policy Optimization (PPO) Schulman et al. (2017) have been introduced which update policies while ensuring monotonic performance improvement. To the best of our knowledge, Log density gradient has only been

discussed in Morimura et al. (2010) in which a TD method to estimate log density gradient for average reward scenarios by using reversible backward Markov chain is proposed.

**Comparison:** In our paper, we re-introduce the idea of log density gradient introduced by Morimura et al. (2010) for estimating gradient in the average reward scenario. Morimura et al. (2010) was also the first work to find out the residual error in policy gradient approximation (Proposition 2). Additionally, this work proposes estimating log density gradient specifically for average reward scenarios ($\gamma = 1$) using a TD update, with additional extensions for linear function approximation.

Our work not only fixes many technical gaps evident in the theory of log density gradient as proposed by Morimura et al. (2010) but also builds on them to make log density gradient practical. We first define log density gradient (equation 10) over a range of discounting factor $\gamma \in [0, 1]$, which also includes the average reward scenario. Using Lemma 2 and 3 we then prove mathematical conditions under which the log density gradient is unique and can be exactly calculated. We further use this relation to propose a TD form of updates (equation 14) for log density gradient estimation for all values of discounting factor $\gamma \in [0, 1]$ as against the average reward scenario proposed by Morimura et al. (2010). In Lemma 4 we further prove that these TD updates converge to a unique solution for all values of discounting factor $\gamma$ except 1. Thus, effectively demonstrating that TD-updates proposed by Morimura et al. (2010) *does not* converge to the true log density gradient, further limiting their use for large scale problems. Additionally, to make log density gradient estimation viable for practical problems, we propose a min-max optimization approach (equation 17) that allows us to estimate log density gradient using empirical samples. We also demonstrate that under linear function approximation settings, this min-max optimization not only has a closed form but also converges to a unique solution (Theorem 1). Under weighted updates, as proposed in algorithm 1 we also show a bound on sample complexity of log density gradient estimation of the order of $\mathcal{O}(\frac{1}{\sqrt{n}})$.

## 3.2 Density Ratio Learning

**Literature survey:** Off-policy evaluation estimates the performance of a target policy $\pi$ using an offline dataset generated by a behavior policy $\mu$ (Voloshin et al., 2019; Katdare et al.). This is done by estimating the average state-action density ratio $\frac{d^\pi_\gamma}{d^\mu}$, which allows approximation of the target policy's performance. In this work, we are primarily interested in the DICE class of off-policy evaluation algorithms (Zhang et al., 2020b; Nachum et al., 2019; Zhang et al., 2020a). These algorithms typically approximate the divergence (for some $f-$divergence of their choice) between the two distributions $d^\pi$ and $d^\mu$ in their convex dual form, eliminating the need to obtain samples from $d^\pi$, which results in a min-max form optimization problem.

**Comparison:** Inspired by the DICE class of algorithms we too propose a min-max form of estimating the log density gradient. We show that such a method of estimating log density gradient converges to the true policy under linear function approximations assumptions. We also show that the sample complexity of such an estimator is of the order $(n^{-1/2})$, with $n$ being the number of on-policy samples.

# 4 Roadmap

Before we begin the technical exposition we would like to give a high level idea of the building blocks involved, and how each step connects to the next.

To that end, we finally propose a min-max version of log density gradient which allows to approximate $\nabla_\theta \log d^{\pi_\theta}$ using empirical samples. Our experimental further show that this manner of policy gradient estimation can potentially make reinforcement learning sample efficient, thus helping them scale.

In Section 5, we first propose a model based approach to approximate the log density gradient for tabular scenarios. To that end, we propose an optimisation problem, the solution to which is the log density gradient. Then we prove some properties about the solution of the optimisation problem, in particular. Next we suggest a temporal difference type method to solve the optimisation problem using samples from the controlled Markov chain. We prove certain uniqueness and convergence properties about the solution obtained from the TD method. However, the TD method works under a very restrctive assumption: it requires backward samples from the Markov chain. To circumvent this assumption, we proceed to the next section.

In Section 6, we first convert the original optimisation problem into a min-max problem using Fenchel duality. Then we propose a TD-type algorithm to solve the min-max problem. This new TD algorithm needs only samples from the forward Markov chain, and subverts the need for backward samples of the previous algorithm. We conclude the section by proving results about the solution of the TD algorithm, and introducing a function approximation framework for the TD method (with proofs of convergence).

## 5 Log Density Gradient

In this section, we introduce log density gradient approach to policy gradient estimation. By deriving the gradient of log density, we demonstrate that traditional policy gradient methods are not well-suited at approximating policy gradient through average reward scenarios. The log density gradient approach addresses this limitation, allowing us to estimate both the average reward and discounted reward formulations. We first propose a model-based approach to log density gradient estimation which relies on Bellman-type recursion to exactly calculate policy gradient. We then extend this to a Temporal difference (TD) version of policy gradient estimation which will allow us to estimate gradient only using on-policy samples.

### 5.1 Model Based Log Density Gradient

The log density gradient method attempts to estimate the gradient of log of average state-action discounted stationery distribution $d_\gamma^{\pi_\theta}$. We start by observing that $d_\gamma^{\pi_\theta}$ satisfies an identity called the Bellman flow equation Liu et al. (2018); Nachum et al. (2019).

**Lemma 1.** *The average state-action density distribution $d_\gamma^{\pi_\theta}$ satisfies the Bellman flow equation $\forall (s, a, s') \times \mathcal{S} \times \mathcal{A} \times \mathcal{S}$,*

$$d_\gamma^{\pi_\theta}(s') = (1 - \gamma)d_0(s') + \gamma \sum_{s,a} d_\gamma^{\pi_\theta}(s,a)\mathcal{P}(s'|s,a) \quad \forall \gamma \in [0,1] \tag{7}$$

A detailed proof for the same can be found in the Appendix 9.3. Note that the Bellmam flow equation 7 equation is similar to the classical Bellman contraction except that it is reversed. Unlike the classical Bellman contraction, the Bellman flow equation shows us the relationship of the next state $s'$ with previous state-action pair $(s, a)$. In that sense, the Bellman flow equation is *reversed* and requires samples from a reversed MDP to estimate $d_\gamma^\pi$ accurately. In the following Lemma, we present an optimisation problem, the solution of which is same as the solution of equation 7. This optimisation problem will be used in later development.

**Lemma 2.** *Choose an arbitrary but fixed $\lambda > 0$. For all $\gamma \in [0,1]$ the solution to the following optimisation is unique and equal to $d_\gamma^{\pi_\theta}$,*

$$\underset{w:\mathcal{S}\to\mathbb{R}}{\arg\min} \sum_{s'} \left( w(s') - (1-\gamma)d_0(s') + \gamma \sum_{s,a} w(s)\pi_\theta(a|s)\mathcal{P}(s'|s,a) \right)^2 + \frac{\lambda}{2}(\sum_s w(s) - 1)^2 \tag{8}$$

Detailed proof can be found in the appendix 9.4. Intuitively speaking, the term $\frac{\lambda}{2}(\sum_s w(s) - 1)^2$ is redundant for $\gamma < 1$, and only becomes useful for average reward scenarios wherein $\gamma = 1$ thus helping ensure uniqueness.

Recall that we are interested in estimating the log density gradient, which is the gradient of the log of density with the policy parameters $\nabla_\theta \log d_\gamma^{\pi_\theta}$. Similar to the Bellman flow equation equation 7, the log density gradient also follows a similar recursion which can be obtained by taking the gradient of the Bellman flow equation 7 with respect to the policy parameters $\theta$ as follows:

$$d_\gamma^{\pi_\theta}(s')\nabla_\theta \log d_\gamma^{\pi_\theta}(s') = \gamma \sum_{s,a} d_\gamma^{\pi_\theta}(s,a)\mathcal{P}(s'|s,a)\nabla_\theta \log d_\gamma^{\pi_\theta}(s,a). \tag{9}$$

Multiplying both sides by $\pi_\theta(a'|s')$ and recalling that $d_\gamma^{\pi_\theta}(s,a) = d_\gamma^{\pi_\theta}(s)\pi_\theta(a|s)$, we obtain

$$d_\gamma^{\pi_\theta}(s',a')(\nabla_\theta \log d_\gamma^{\pi_\theta}(s',a') - \nabla_\theta \log \pi(a'|s')) = \gamma \sum_{s,a} d_\gamma^{\pi_\theta}(s,a)\nabla_\theta \log d_\gamma^{\pi_\theta}(s,a)\mathcal{P}(s'|s,a)\pi_\theta(a'|s') \tag{10}$$

Furthermore, if we have exact knowledge of the transition matrix $\mathcal{P}$, we can exactly calculate the log density gradient by solving for the following optimization problem

$$\min_{w:\mathcal{S}\times\mathcal{A}\to\mathbb{R}^n}\left\{\mathbb{E}_{(s',a')\sim d_\gamma^{\pi_\theta}}\|\nu(s',a')\|^2 + \frac{\lambda}{2}\left\|E_{(s,a)\sim d_\gamma^{\pi_\theta}}[w(s',a')]\right\|^2\right\} \tag{11}$$

$$\nu(s',a') := d_\gamma^{\pi_\theta}(s',a')(w(s',a') - \nabla_\theta \log \pi_\theta(a'|s')) - \gamma \sum_{s,a} d_\gamma^{\pi_\theta}(s,a)\mathcal{P}(s'|s,a)\pi_\theta(a'|s')w(s,a)$$

**Lemma 3.** *For $\lambda > 0$, the solution to equation 11 is unique and equal to $\nabla_\theta \log d_\gamma^{\pi_\theta}$ for all $\gamma \in [0,1]$.*

We describe the proof for this Lemma in the appendix 9.5. Note that the constraint $\frac{\lambda}{2}\|E_{(s,a)\sim d_\gamma^{\pi_\theta}}[w(s',a')]\|^2$ is redundant for $\gamma < 1$ and only becomes useful for average reward scenarios $\gamma = 1$. For a finite state and action space, the proof follows from the fact that the optimal solution to equation 11 requires solving for a linear equation $A \cdot w = b$. The remaining part of the proof demonstrates that this matrix $A$ is invertible for $\forall \gamma \in [0,1]$. It is worth reiterating that, once we have an estimate $\nabla_\theta \log d_\gamma^{\pi_\theta}$, we can use this estimate to approximate the policy gradient using equation 6. We will now recall two important properties of log density gradient.

**Proposition 1.** *The policy gradient method as recalled from equation 6*

$$\nabla_\theta J_\gamma(\pi) = \mathbb{E}_{(s,a)\sim d_\gamma^{\pi_\theta}}[\nabla_\theta \log d_\gamma^{\pi_\theta}(s,a) \cdot r(s,a)].$$

*is exactly equal to the classical policy gradient (Sutton et al., 1999) recalled from equation 2*

$$\nabla_\theta J_\gamma(\pi_\theta) = \mathbb{E}_{(s,a)\sim d_\gamma^{\pi_\theta}}[Q_\gamma^{\pi_\theta}(s,a) \cdot \nabla_\theta \log \pi_\theta(a|s)], \quad \gamma \in [0,1].$$

Detailed proof for this proposition can be found in Appendix 9.2. In essence this results shows us that log density gradient approach to policy gradient estimates the exact same gradient but using a different formulation. Additionally log density gradient formulation allows us to estimate policy gradient for average reward formulations. It is well known that, the classical policy gradient theorem is in-sufficient to estimate gradient for average reward scenarios. This is because the Bellman equation for reward case ($\gamma = 1$) is not a contraction, making it harder to approximate Q-function using neural networks. As a compromise, reinforcement learning practioners typically approximate policy gradient for average reward scenario using a discounting factor slightly less than 1 and use that to update policy. Our next result shows that such a way to approximating policy gradient ignores a residual error, which is not necessarily small. Correcting for this residual error, can not improve policy gradient estimation but also potentially help improve sample complexity of reinforcement learning algorithms.

**Proposition 2.** *The following identity (recalled from equation 5) is true*

$$\nabla_\theta J_1(\pi_\theta) = \mathbb{E}_{(s,a)\sim d_1^{\pi_\theta}}[\textcolor{red}{Q_\gamma^{\pi_\theta}(s,a)} \cdot \nabla_\theta \log \pi_\theta(a|s)] + \underbrace{(1-\gamma)\mathbb{E}_{(s,a)\sim d_1^{\pi_\theta}}[\nabla_\theta \log d_1^{\pi_\theta}(s) \cdot V_\gamma^{\pi_\theta}(s)]}_{\textit{Residual Error}}$$

*Proof.* From the definition of log density gradient equation 6 we have $\nabla_\theta J_1(\pi_\theta) = \mathbb{E}_{(s,a)\sim d_1^{\pi_\theta}}[\nabla_\theta \log d_1^{\pi_\theta}(s,a) \cdot r(s,a)]$. Let $\gamma < 1$, and we use the Bellman equation 1b to obtain

$$\nabla_\theta J_1(\pi_\theta) = \mathbb{E}_{(s,a)\sim d_1^{\pi_\theta}}[\nabla_\theta \log d_1^{\pi_\theta}(s,a) \cdot (Q_\gamma^{\pi_\theta}(s,a) - \gamma\mathbb{E}_{s'\sim\mathcal{P}(\cdot|s,a)}[V_\gamma^{\pi_\theta}(s')])] \tag{12a}$$

$$= \mathbb{E}_{(s,a)\sim d_1^{\pi_\theta}}[(\nabla_\theta \log d_1^{\pi_\theta}(s) + \nabla_\theta \log \pi_\theta(a|s)) \cdot (Q_\gamma^{\pi_\theta}(s,a) - \gamma\mathbb{E}_{s'\sim\mathcal{P}(\cdot|s,a)}[V_\gamma^{\pi_\theta}(s')])] \tag{12b}$$

$$= \mathbb{E}_{(s,a)\sim d_1^{\pi_\theta}}[Q_\gamma^{\pi_\theta}(s,a) \cdot \nabla_\theta \log \pi_\theta(a|s)] + (1-\gamma)\mathbb{E}_{s\sim d_1^{\pi_\theta}}[\nabla_\theta \log d_1^{\pi_\theta}(s) \cdot V^{\pi_\theta}(s)] \tag{12c}$$

Here, we go from equation 12a to 12b by utilizing $\nabla_\theta \log d_\gamma^{\pi_\theta}(s,a) = \nabla_\theta \log d_\gamma^{\pi_\theta}(s) + \nabla_\theta \log \pi_\theta(a|s)$. We finally go from 12b to 12c by using a key identity of log density gradient equation 10.

Note that the first term in equation 12b is the practical instantiation of classical policy gradient theorem, while the second term $(1-\gamma)\mathbb{E}_{s\sim d_1^{\pi_\theta}}[\nabla_\theta \log d_1^{\pi_\theta}(s) \cdot V_\gamma^{\pi_\theta}(s)]$ being the residual error. This completes the proof. $\qquad\square$

To recap, in this section, we propose a Bellman-like relationship for the log density gradient, which further allows us estimate the gradient using a simple but known optimization. Next, we propose a Temporal difference (TD) type method to estimate log density gradient using samples.

### 5.2 Temporal Difference Log Density Gradient

In this section, we begin to estimate log density gradient algorithm using just on-policy samples. To that end, as a first step, we propose a temporal difference approach to log density gradient estimation. We refer to our method as TD(0) method. [1] To get an update equation for our TD(0) method, consider a re-arranged version of equation 10 for log density gradient,

$$\nabla_\theta \log d_\gamma^{\pi_\theta}(s', a') = \nabla_\theta \log \pi_\theta(a'|s') + \gamma \sum_{s,a} \frac{d_\gamma^{\pi_\theta}(s,a)\mathcal{P}(s'|s,a)\pi_\theta(a'|s')}{d_\gamma^{\pi_\theta}(s',a')} \nabla_\theta \log d_\gamma^{\pi_\theta}(s,a) \tag{13}$$

As mentioned earlier, this recursion is backwards as compared to the Bellman contraction. In order to see that, we will first define the backwards distribution $\mathcal{P}_b : \mathcal{S} \times \mathcal{A} \to \Delta(\mathcal{S}, \mathcal{A})$ such that

$$\mathcal{P}_b(s,a|s',a') := \frac{d_\gamma^{\pi_\theta}(s,a)\mathcal{P}(s'|s,a)\pi_\theta(a'|s')}{d_\gamma^{\pi_\theta}(s',a')} = \frac{d_\gamma^{\pi_\theta}(s,a)\mathcal{P}^{\pi_\theta}(s',a'|s,a)}{d_\gamma^{\pi_\theta}(s',a')}.$$

This backwards distribution $\mathcal{P}_b$ estimates the probability of the previous state $(s, a)$ given the next state $(s', a')$. The mathematical form of the backwards distribution is thus, consequence of Bayes' rule. The summation in equation 13 therefore becomes an expectation under $\mathcal{P}_b$

$$\nabla_\theta \log d_\gamma^{\pi_\theta}(s', a') = \nabla_\theta \log \pi_\theta(a'|s') + \gamma \mathbb{E}_{(s,a)\sim\mathcal{P}_b(\cdot|s,a)}[\log d_\gamma^{\pi_\theta}(s,a)].$$

The log density gradient is therefore said to follow a backward recursion and it requires samples from backward conditional probability $\mathcal{P}_b$ to estimate log density gradient[2]. Assuming that we have samples from this backward distribution $\mathcal{P}_b$ we now propose TD updates to estimate log density gradient $w$ for all discounting factor $\gamma \in [0, 1]$,

$$w(s', a') \leftarrow w(s', a') + \alpha[\gamma w(s, a) + g(s', a') - w(s', a')], \tag{14}$$

with $(s', a') \sim d_\gamma^{\pi_\theta}, (s, a) \sim \mathcal{P}_b(\cdot|s', a')$ and $g(s', a') := \nabla_\theta \log \pi_\theta(a'|s')$. It is worth noting that this TD(0) update is a generalization of Morimura et al. (2010), who only propose these updates for average reward scenario ($\gamma = 1$).

Next we define an operator $Y_\gamma$ to capture the behaviour of the update rule equation 14 after taking expectation as,

$$(Y_\gamma \cdot w)(s', a') := \gamma \mathbb{E}_{(s,a)\sim\mathcal{P}_b(\cdot|s',a')}[w(s, a)] + g(s', a').$$

We can write this in matrix form as follows,

$$Y_\gamma \cdot W = \gamma D_{\pi_\theta}^{-1} \mathcal{P}_{\pi_\theta}^\top D_{\pi_\theta} W + G \tag{15}$$

where, $W \in \mathbb{R}^{|\mathcal{S}|\cdot|\mathcal{A}|\times n}$ is the matrix with every row corresponding to $w(s, a)$ for each state-action pair $(s, a)$. Similarly, $G \in \mathbb{R}^{|\mathcal{S}|\cdot|\mathcal{A}|\times n}$ has its rows as $\nabla_\theta \log \pi_\theta$ for each state-action pair. Let $\mathcal{P}_{\pi_\theta}, D_{\pi_\theta} \in \mathbb{R}^{|\mathcal{S}|\cdot|\mathcal{A}|\times|\mathcal{S}|\cdot|\mathcal{A}|}$ where $(\mathcal{P}_{\pi_\theta})_{((s,a),(s',a'))} = \mathbb{P}^{\pi_\theta}(s', a'|s, a)$ and $D_{\pi_\theta}$ is a diagonal matrix whose every element correspond to $d_\gamma^{\pi_\theta}$ for each state-action pair. We use this matrix form for the operator $Y_\gamma$ in the proof of the following lemma.

**Lemma 4.** *Let $w_0 \in \Delta(\mathcal{S}, \mathcal{A})$ be an arbitrary initial guess. Let $w_k = Y_\gamma \cdot w_{k-1}$ for all natural numbers $k \geq 1$. For $\gamma \in [0, 1)$, the operator $Y_\gamma$ is a contraction, and $\{w_k\}_{k\geq0}$ converges to a unique fixed point $\nabla_\theta \log d_\gamma^{\pi_\theta}$.*

---

[1]There is a family of TD methods which incorporate prior traces in a discounted manner called TD($\lambda$) (Tesauro, 1992).
[2]Although for $\gamma = 1$ we can use samples from $\mathcal{P}$ as well (Morimura et al., 2010)

Detailed proof of Lemma 4 can be found in Appendix 9.6. The key idea behind this Lemma is to show that successive application of the above mentioned TD updates will help us estimate the log density gradient accurately. In-fact we also show that these TD updates are a contraction, thus effectively ensuring convergence. Extension of Lemma 4 to linear function approximation, and proof of convergence for the same, can be found in the Appendix 9.7.

Although TD methods are known to converge, they still suffer from two problems. One, the access to samples from backward conditional probability. Two, scalability to large problem spaces. We attempt to solve both of these problems in the next section where we propose a min-max optimization procedure for estimating the log density gradient.

## 6 Min-Max Log Density Gradient

In this section, we propose min-max optimization approach to evaluate log density gradient for all values of the discounting factor including the average reward scenario ($\gamma = 1$). Doing so removes the need for samples from backward distribution, which was limiting scalability of log density gradient method described in the previous section. Min-max optimizations also allow us to use a large variety of function classes like neural networks to approximate log density gradient.

Let us return to the loss function that we initially propose in equation 11. Classical machine learning algorithms usually posit loss function as Empiricial Risk Minimization (ERM), which allows us to approximate the loss using samples from that distribution. In order to bring our key optimization in an ERM form, consider a modified form of the optimization proposed in equation 11 (the modification is that $\delta(s', a')$ is divided by $d_\gamma^{\pi_\theta}(s', a')$ where the ergodicity assumption ensures this operation is well defined),

$$\underset{w \in \mathcal{S} \times \mathcal{A} \to \mathbb{R}^n}{\arg\min} \ \mathbb{E}_{(s',a') \sim d_\gamma^{\pi_\theta}} \left[ \left\| \frac{\nu(s', a')}{d_\gamma^{\pi_\theta}(s', a')} \right\|^2 \right] + \frac{\lambda}{2} \| \mathbb{E}_{(s,a) \sim d_\gamma^{\pi_\theta}}[w(s, a)] \|^2 \tag{16}$$

$$\nu(s', a') := d_\gamma^{\pi_\theta}(s', a')(w(s', a') - \nabla_\theta \log \pi_\theta(a'|s')) - \gamma \sum_{s,a} d_\gamma^{\pi_\theta}(s, a) \mathcal{P}(s'|s, a) \pi(a'|s') w(s, a)$$

The denominator term $d_\gamma^{\pi_\theta}$ is added to ensure that the final optimization form can be written in form of an expectation, which we shall see soon. This allows us to use samples to approximate our optimization function. We also add the regularization term $\frac{\lambda}{2} \| \mathbb{E}_{(s,a) \sim d_\gamma^{\pi_\theta}}[w(s, a)] \|^2$ which allows us to estimate log density gradient even for all values of the discounting factor including average reward scenarios ($\gamma = 1$).

Since, equation equation 16 is a re-weighting of equation 11 with the $(d_\gamma^{\pi_\theta}(s, a))^{-1}$. the optimal solution for the both the equation is the same, thus ensuring uniqueness of the optimization.

By exploiting the Fenchel-duality, we can re-write this optimization in the minimax form (Rockafellar, 2015; Zhang et al., 2020b) as follows,.

$$\arg \min_{w:\mathcal{S} \times \mathcal{A} \to \mathbb{R}^d} \max_{f:\mathcal{S} \times \mathcal{A} \to \mathbb{R}^d, \tau \in \mathbb{R}^d} L_\gamma(w, f, \tau) := \left\{ \mathbb{E}_{(s',a') \sim d_\gamma^{\pi_\theta}}[f(s', a') \cdot w(s', a')] \right.$$

$$- \mathbb{E}_{(s',a') \sim d_\gamma^{\pi_\theta}}[f(s', a') \cdot \nabla_\theta \log \pi_\theta(a'|s')] - \gamma \mathbb{E}_{(s,a) \sim d_\gamma^{\pi_\theta}}[\mathbb{E}_{s' \sim \mathcal{P}(\cdot|s,a), a' \sim \pi_\theta(\cdot|s')}[f(s', a')] \cdot w(s, a)]$$

$$\left. - \frac{1}{2} \mathbb{E}_{(s,a) \sim d_\gamma^{\pi_\theta}}[\|f(s', a')\|^2] + \lambda(\tau \cdot \mathbb{E}_{(s,a) \sim d_\gamma^{\pi_\theta}}[w(s, a)] - \frac{1}{2} \|\tau\|^2) \right\} \tag{17}$$

In many cases searching over all function mappings $\mathcal{S} \times \mathcal{A} \to \mathbb{R}^d$ is not possible, hence we search over a smaller and more tractable function classes $\mathcal{W}, \mathcal{F}$ and the aim is to approximate

$$\nabla_\theta \log d_\gamma^{\pi_\theta} \approx \arg \min_{w \in \mathcal{W}} \max_{f \in \mathcal{F}, \tau \in \mathbb{R}^n} L_\gamma(w, f, \tau).$$

Such a practical consideration allows us to use different types of function approximators like linear function approximation, neural networks, and reproducible kernel Hilbert spaces (RKHS).

In the remainder of this section, we will focus on linear function approximation, and provide an update rule to solve equation 17 under linear function approximation. For that we choose a feature map $\Phi : \mathcal{S} \times \mathcal{A} \to \mathbb{R}^d$ and parameters $\alpha, \beta \in \mathbb{R}^{d \times n}$ that need to be learnt, so that we can approximate the optima of equation 17, $w^*(s, a)$ and $f^*(s, a)$ with $\alpha^T \Phi(s, a)$, and $\beta^T \Phi(s, a)$ respectively, for each state action pair $(s, a)$. The update rule is

$$\delta_t = \Phi_t \Phi_t^T - \gamma \Phi_t (\Phi_t')^T \tag{18a}$$

$$\alpha_{t+1}^T = \alpha_t^T - \varepsilon_t (\beta^T \delta_t + \lambda(\tau \Phi_t^T)) \tag{18b}$$

$$\beta_{t+1}^T = \beta_t^T + \varepsilon_t (\alpha_t^T \delta_t - g_t \Phi_t^T - \beta_t^T \Phi_t \Phi_t^T) \tag{18c}$$

$$\tau_{t+1} = \tau_t + \varepsilon_t (\lambda(\alpha_t^T \Phi_t - \tau_t)) \tag{18d}$$

where, $\Phi_t := \Phi(s_t, a_t)$ is the feature encountered at time $t$, $g_t := \nabla_\theta \log \pi_\theta(a_t | s_t)$, and $\Phi_t' := \Phi_t(s_t', a_t')$ for $(s_t, a_t) \sim d_\gamma^{\pi_\theta}, s_t' \sim \mathcal{P}(\cdot | s_t, a_t), a_t' \sim \pi_\theta(a_t' | s_t')$. We first re-write the updates in equation 18 in form of $d_t = [\alpha_t, \beta_t, \tau_t^T]$ so that the updates can be written in matrix form $d_{t+1} = d_t + \varepsilon_t(G_{t+1} d_t + h_{t+1})$, where, $G_{t+1}, h_{t+1}$ are as follows,

$$G_{t+1} := \begin{bmatrix} 0 & -A_t & -\lambda\Phi_t \\ A_t & -C_t & 0 \\ \lambda\Phi_t^T & 0 & -\lambda \end{bmatrix}, \quad h_{t+1} := \begin{bmatrix} 0 \\ -B_t \\ 0 \end{bmatrix}$$

and $A_t := (\Phi_t \Phi_t^T - \gamma \Phi_t (\Phi_t')^T), B_t := \Phi_t g_t^T, C_t := \Phi_t \Phi_t^T$. We can calculate the expectation for each of these matrices as follows,

$$G := \mathbb{E}_p[G_{t+1}] = \begin{bmatrix} 0 & -A & -\lambda\Psi D_{\pi_\theta} e \\ A & C & 0 \\ \lambda e^T D_{\pi_\theta} \Psi^T & 0 & -\lambda \end{bmatrix}, \quad h := \mathbb{E}_{(s,a) \sim d_\gamma^{\pi_\theta}}[h_{t+1}] = \begin{bmatrix} 0 \\ -B \\ 0 \end{bmatrix}$$

Here, each column of $\Psi \in \mathbb{R}^{|\mathcal{S}| \cdot |\mathcal{A}| \times n}$ is the feature vector $\Phi(s, a)$, for each $(s, a) \in \mathcal{S} \times \mathcal{A}$ and $e \in \mathbb{R}^n$ is a vector of 1's at every element. We can similarly write $A = \Psi D_{\pi_\theta}(I - \gamma P_{\pi_\theta})\Psi^T, B = \Psi D_{\pi_\theta} G^T, C = \Psi D_{\pi_\theta} \Psi^T$ and $E_p[\cdot] := \mathbb{E}_{(s,a) \sim d_\gamma^{\pi_\theta}, s' \sim \mathcal{P}(\cdot | s, a), a' \sim \pi_\theta(\cdot | s')}[\cdot]$. We can now prove the convergence of linear function approximation under the following key assumptions.

**Assumption 1.** *1. The matrix $\Psi$ has linearly independent columns.*

*2. The matrix $A$ is non-singular or the regularizer $\lambda > 0$.*

*3. The feature matrix $\Phi$ has uniformly bounded second moments.*

**Theorem 1.** *Under the assumptions 1, the update equation 18 converges in probability to a unique solution. That is, $\lim_{t \to \infty} d_t = G^{-1} h$ in probability.*

The detailed proof is provided in Appendix 9.8. The proof is similar to (Zhang et al., 2020b, Theorem 2) and invokes theorem 2.2 Borkar & Meyn (2000).

We provide a sample complexity analysis for a projected version of the update rule equation 18. To that end, we propose Algorithm 1 called the Projected Log Density Gradient. We choose closed, bounded and convex sets $X \subset \mathbb{R}^{d \times n}, Y \subset \mathbb{R}^{d \times n}, Z \subset \mathbb{R}^{1 \times n}$ and define a projection operator $\Pi_X, \Pi_Y, \Pi_Z$ that project our variables $\alpha_t, \beta_t, \tau_t$ onto $X, Y, Z$ respectively. Moreover, we choose a learning rate $\{\varepsilon_t\}_{t=1}^m$ where we run the algorithm for $m$ steps. The details of the choice of learning rate are found in Appendix 9.9.

**Theorem 2.** *Under assumptions 1 for $(\bar{\alpha}, \bar{\beta}, \bar{\tau})$ obtained from Algorithm 1 after $m$ steps, the optimality gap $\epsilon_g(\bar{\alpha}\bar{\beta}, \bar{\tau})$ (defined below) is bounded with probability $1 - \delta$ as follows,*

$$\epsilon_g(\bar{\alpha}, \bar{\beta}, \bar{\tau}) := \max_{(\beta,\tau) \in Y \times Z} L(\bar{\alpha}, \beta, \tau) - \min_{\alpha \in X} L(\alpha, \bar{\beta}, \bar{\tau}) \leq C_0 \sqrt{\frac{5}{m}} (8 + 2 \log \frac{2}{\delta}) \quad w.p. \ 1 - \delta$$

*where, $C_0$ is a constant which is a function of the sets $X, Y, Z$, and the second moment of $\Phi$.*

We present the proof of this result in appendix 9.9. This result essentially shows us that the upper-bound for log density gradient estimation requires $O(\frac{1}{\sqrt{m}})$ (where $m$ is the number of steps the algorithm runs for) samples to learn an accurate estimation.

---

**Algorithm 1** Projected Log Density Gradient

---

1: **for** $t = 1, 2, ..., m$ **do:**
2: $\delta_t = \Phi_t \Phi_t^T - \gamma \Phi_t (\Phi_t')^T$
3: $\alpha_{t+1}^T = \Pi_X (\alpha_t^T - \varepsilon_t (\beta^T \delta_t + \lambda(\tau \Phi_t^T)))$
4: $\beta_{t+1}^T = \Pi_Y (\beta_t^T + \varepsilon_t (\alpha_t^T \delta_t - g_t \Phi_t^T - \beta_t^T \Phi_t \Phi_t^T))$
5: $\tau_{t+1} = \Pi_Z (\tau_t + \varepsilon_t (\lambda(\alpha_t^T \Phi_t - \tau_t)))$
6: **Return** $\bar{\alpha}, \bar{\beta}, \bar{\tau}$
   Where, $\bar{\alpha} = \frac{\sum_{i=1}^n \varepsilon_i \alpha_i}{\sum_{i=0}^n \varepsilon_i}$, $\bar{\beta} = \frac{\sum_{i=1}^n \varepsilon_i \beta_i}{\sum_{i=0}^n \varepsilon_i}$, $\bar{\tau} = \frac{\sum_{i=1}^n \varepsilon_i \tau_i}{\sum_{i=0}^n \varepsilon_i}$

---

**Algorithm 2** Linear Log Density Gradient

---

1: **for** $t = 1, 2, ..., m$ **do:**
2: $\alpha^* = \arg \min_{\alpha^\cdot \in \mathbb{R}^{d \times n}} \max_{\beta: \in \mathbb{R}^{d \times n}, \tau \in \mathbb{R}^{1 \times n}} L_\gamma(w, f, \tau)$ (equation 17)
3: $\nabla_\theta J_1(\pi_\theta) = \mathbb{E}_{(s,a) \sim d_1^{\pi_\theta}} [\nabla_\theta \log d_1^{\pi_\theta}(s,a) \cdot r(s,a)]$
4: **Return** $\theta$

---

## 7 Experiments

In this section, we present a proof of concept for our log density gradient estimation on two sets of environments $5 \times 5$ and $3 \times 3$ gridworld environment (Towers et al., 2023). For the gridworld experiments, we approximate log density gradient by using linear function approximation (Algorithm 2). Here, the features are $\phi : \mathcal{S} \times \mathcal{A} \to \mathbb{R}^{|\mathcal{S}| \cdot |\mathcal{A}|}$ such that it maps every state to the corresponding standard basis vector. Our results for $5 \times 5$ are in Figure 2 and for $3 \times 3$ in Figure 3.

We compare our algorithm against 3 different baselines. The first is theoretical log density gradient as described in Lemma 3. The second baseline implements REINFORCE algorithm, which is the practical rendition of the policy gradient theorem (Williams, 1992). The third is theoretical policy gradient method which exactly computes the classical policy gradient theorem, as in equation 4b (Sutton et al., 1999).

We observe in that both log density gradient approaches are more sample efficient than both policy gradient approaches. This is because policy gradient methods approximate the gradient for average reward scenarios $(\gamma = 1)$ by estimating a Q-function for a discounting factor less than 1. Moreover, we observe that our method tends to outperform REINFORCE with much reduced variance. Our approach is always very close in performance to the theoretical log density gradient which serves to validate correctness of our algorithm. In $5 \times 5$ gridworld we also observe our algorithm to outperforms theoretical log density gradient. This is because, theoretical log density gradient suffers from some numerical computation issues arising from average reward scenarios.

## 8 Conclusion and Future Work

We present log density gradient algorithm that estimates policy gradient using state-action discounted formulation of a reinforcement learning problem. We observe that policy gradient estimated in this manner, corrects for a residual error common in many reinforcement learning tasks. We show that with a known model, we can exactly calculate the gradient of the log density by solving two sets of linear equations. We further propose a TD(0) algorithm to implement the same, but it needs samples from the backward Markov chain, which becomes too restrictive. Therefore, we propose a min-max optimization that estimates log density gradient using just on-policy samples. We not only prove theoretical properties like convergence and uniqueness but also experimentally demonstrate that our method is sample efficient as compared to classical policy gradient methods like REINFORCE. This approach looks promising, and further studies of log density gradient will focus on scaling their performance to complex tasks.
**Limitations:** Currently most of our experimental results require Linear function approximation to estimate log density gradient. This is because under linear function approximation conditions, our min-max optimization equation 17 becomes a quadratic program and thus has a closed form solution. To scale log density gradient

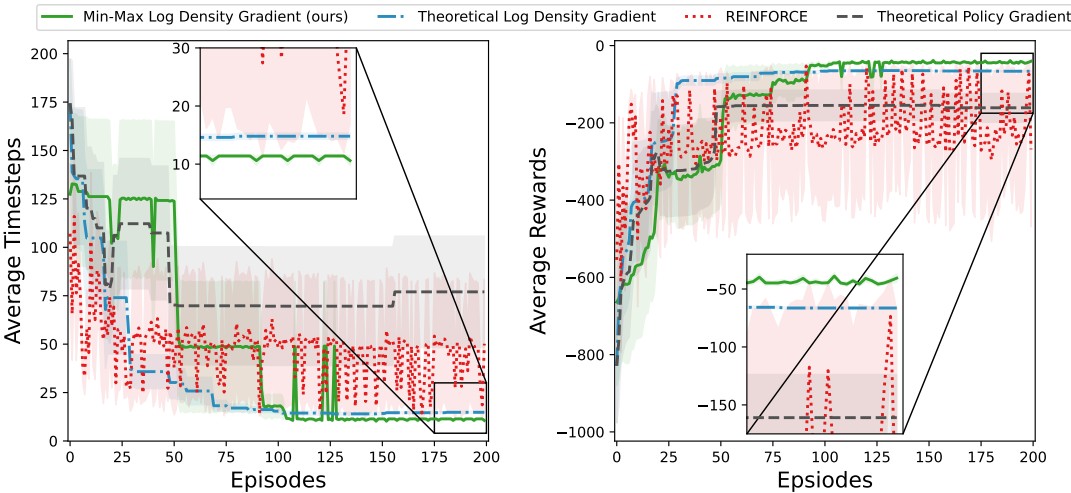

Figure 2: For $5 \times 5$ gridworld, comparison of Log Density Gradient algorithms (in light green) as compared to REINFORCE (light red), theoretical policy gradient (gray) and theoretical log density gradient (blue). We observe that our empirical algorithm comfortably outperforms the other baselines.

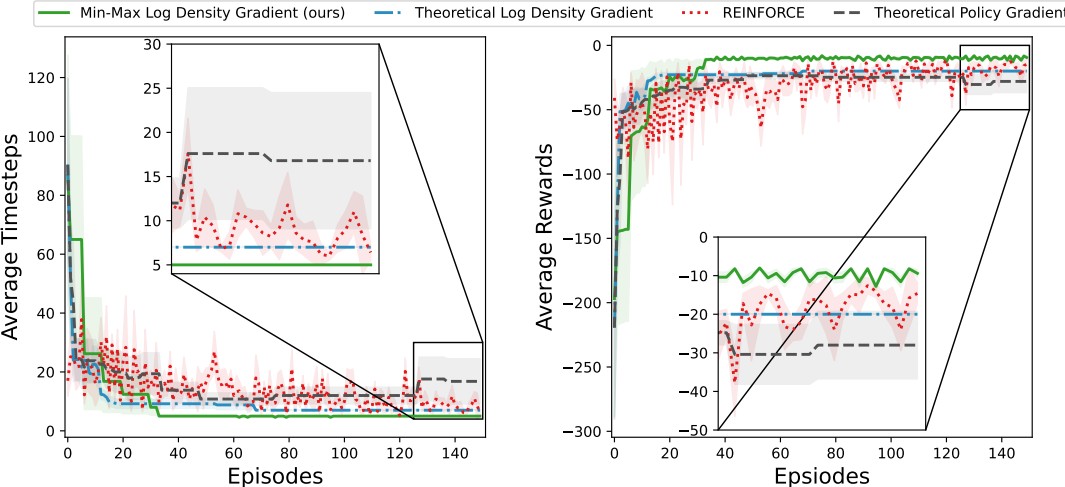

Figure 3: For $3 \times 3$ gridworld, comparison of Log Density Gradient algorithms (in light green) as compared to REINFORCE (light red), theoretical policy gradient (gray) and theoretical log density gradient (blue). We observe that our empirical algorithm comfortably outperforms the other baselines.

algorithms to complex cases requires evaluating high quality features which will also allow us to estimate gradient accurately Zhang et al. (2022).

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
