# 9 Appendix

## 9.1 Additional Literature Review - Temporal Difference Methods

Temporal difference (TD) learning involves improving predictions through bootstrapping the current estimation. Early TD methods were mainly used to estimate the value function $V^{\pi_\theta}$, which used a semi-gradient update rule to improve value function prediction (Sutton, 1988). More advanced TD methods incorporate prior traces (TD($\lambda$)) in a discounted manner (Tesauro, 1992). Since many reinforcement learning methods are not tabular, Linear TD methods were also proposed that uses this methodology to learn parameters that approximate the value function (Bradtke & Barto, 1996; Boyan, 1999). Recently, TD methods have been effectively used to approximate Advantage function (Mnih et al., 2016), density ratio function (Gelada & Bellemare, 2019; Hallak & Mannor, 2017; Gelada & Bellemare, 2019) and off-policy value estimation (Maei, 2011).In this work, we will observe that the log density gradient has a recursive form that cannot be approximated using a closed form solution. We propose a TD(0) methods to estimate the log density gradient using on-policy samples.

## 9.2 Proof of Proposition 1

*Proof.* We begin with policy gradient calculated using log density gradient, equation 6)

$$\nabla_\theta J_\gamma(\pi_\theta) = \mathbb{E}_{(s,a)\sim d_\gamma^{\pi_\theta}}[\nabla_\theta \log d_\gamma^{\pi_\theta}(s,a) r(s,a)]$$

We recall from Bellman equation, 1b that $r(s,a) = Q_\gamma^{\pi_\theta} - \gamma \mathbb{E}_{s'\sim\mathcal{P}(\cdot|s,a),a'\sim\pi_\theta(\cdot|s')}[Q_\gamma^{\pi_\theta}(s',a')]$, hence

$$\nabla_\theta J_\gamma(\pi_\theta) = \mathbb{E}_{(s,a)\sim d_\gamma^{\pi_\theta}}[\nabla_\theta \log d_\gamma^{\pi_\theta}(s,a)(Q_\gamma^{\pi_\theta}(s,a) - \gamma \mathbb{E}_{s'\sim\mathcal{P}(\cdot|s,a)\pi_\theta(a'|s')}[Q_\gamma^{\pi_\theta}(s',a')])$$

We recover the policy gradient theorem in equation 2 by first multiplying equation 10 by $Q_\gamma^{\pi_\theta}(s',a')$

$$d_\gamma^{\pi_\theta}(s',a')(\nabla_\theta \log d_\gamma^{\pi_\theta}(s',a') - \nabla_\theta \log \pi(a'|s'))Q_\gamma^{\pi_\theta}(s',a')$$
$$= \gamma \sum_{s,a} d_\gamma^{\pi_\theta}(s,a)\nabla_\theta \log d_\gamma^{\pi_\theta}(s,a)\mathcal{P}(s'|s,a)\pi_\theta(a'|s')Q_\gamma^{\pi_\theta}(s',a')$$

and then re-arranging terms with $\nabla_\theta \log d_\gamma^\pi$ on the left hand side and the remaining terms on the right hand side. We can then sum these terms to get equation 2

$$\nabla_\theta J_\gamma(\pi_\theta) = \mathbb{E}_{(s,a)\sim d_\gamma^{\pi_\theta}}[\nabla_\theta \log d_\gamma^{\pi_\theta}(s,a)(Q_\gamma^{\pi_\theta}(s,a) - \gamma \mathbb{E}_{s'\sim\mathcal{P}(\cdot|s,a)\pi_\theta(a'|s')}[Q_\gamma^{\pi_\theta}(s',a')])$$
$$= \mathbb{E}_{(s,a)\sim d_\gamma^{\pi_\theta}}[Q_\gamma^{\pi_\theta}(s,a)\nabla_\theta \log \pi(a|s)]$$

This completes the proof. $\qquad\square$

## 9.3 Proof of Lemma 1

*Proof.* Recall the definition of $d_\gamma^{\pi_\theta}$ from equation 3

$$d_\gamma^{\pi_\theta}(s) = (1-\gamma)\sum_{t=0}^{\infty}\gamma^t \mathbb{P}(s_t = s|s_0 \sim d_0, a_t \sim \pi_\theta(s_t), s_{t+1} \sim \mathcal{P}(s_t, a_t))$$

We see that for a given state $s' \in \mathcal{S}$ the occupancy of an agent at time $t$ and $t+1$ are related as follows,

$$\mathbb{P}(s_{t+1} = s') = \sum_{s\in\mathcal{S},a\in\mathcal{A}} \mathbb{P}(s_t = s)\cdot\pi_\theta(a|s)\cdot\mathcal{P}(s'|s,a)$$

If we multiply both sides by $\gamma^{t+1}$ and sum them up from $t = 0$ to $\infty$. We get the following form,

$$\sum_{t=0}^{\infty} \gamma^{t+1} \mathbb{P}(s_{t+1} = s') = \gamma \sum_{t=0}^{\infty} \gamma^t \sum_{s \in \mathcal{S}, a \in \mathcal{A}} \mathbb{P}(s_t = s)\pi_\theta(a|s)\mathcal{P}(s'|s, a)$$

$$= \gamma \sum_{s \in \mathcal{S}, a \in \mathcal{A}} \sum_{t=0}^{\infty} \gamma^t \mathbb{P}(s_t = s)\pi_\theta(a|s)\mathcal{P}(s'|s, a)$$

$$= \frac{\gamma}{1 - \gamma} \sum_{s \in \mathcal{S}, a \in \mathcal{A}} d_\gamma^{\pi_\theta}(s)\pi_\theta(a|s)\mathcal{P}(s'|s, a)$$

We go from the first equation to the second by exchanging the summation signs and from the second to the third by using the definition of $\sum_{t=0}^{\infty} \gamma^t \mathbb{P}(s_t = s) = d_\gamma^{\pi_\theta}$. We add $\mathbb{P}(s_0 = s')$ on both sides, which is nothing but the set of initial states $d_0(s')$, to get

$$\sum_{t=0}^{\infty} \gamma^t \mathbb{P}(s_t = s') = d_0(s') + \frac{\gamma}{1 - \gamma} \sum_{s \in \mathcal{S}, a \in \mathcal{A}} d_\gamma^{\pi_\theta}(s)\pi_\theta(a|s)\mathcal{P}(s'|s, a)$$

$$\implies \frac{1}{1 - \gamma} d_\gamma^{\pi_\theta}(s') = d_0(s') + \frac{\gamma}{1 - \gamma} \sum_{s \in \mathcal{S}, a \in \mathcal{A}} d_\gamma^{\pi_\theta}(s)\pi_\theta(a|s)\mathcal{P}(s'|s, a)$$

$$\implies d_\gamma^{\pi_\theta}(s') = (1 - \gamma)d_0(s') + \gamma \sum_{s \in \mathcal{S}, a \in \mathcal{A}} d_\gamma^{\pi_\theta}(s)\pi_\theta(a|s)\mathcal{P}(s'|s, a)$$

where we used

$$\sum_{t=0}^{\infty} \gamma^{t+1} \mathbb{P}(s_{t+1} = s') + d_0(s') = \sum_{t=0}^{\infty} \gamma^t \mathbb{P}(s_t = s') = \frac{1}{1 - \gamma} d_\gamma^{\pi_\theta}(s').$$

This completes the proof. □

### 9.4 Proof of Lemma 2

*Proof.* Recall the optimization 8

$$\arg\min_{w:\mathcal{S} \to \mathbb{R}} \sum_{s'} (w(s') - (1 - \gamma)d_0(s') + \gamma \sum_{s,a} w(s)\pi_\theta(a|s)\mathcal{P}(s'|s, a))^2 + \frac{\lambda}{2}(\sum_s w(s) - 1)^2$$

It is worth noting that the optimization has two quadratic terms. Thus, the lowest value that they can take is only when $w(s') - (1 - \gamma)d_0(s') + \gamma \sum_{s,a} w(s)\pi_\theta(a|s)\mathcal{P}(s'|s, a) = 0$ and $\sum_s w(s) = 1$. For the next part of the proof, we will handle them case-by-case.

**Case 1.** $\gamma < 1$

Recall that the first term of the optimization equation 8 goes to zero only when equation 7 (reproduced below) is satisfied

$$w(s') = (1 - \gamma)d_0(s') + \gamma \sum_{s \in \mathcal{S}, a \in \mathcal{A}} w(s)\pi_\theta(a|s)\mathcal{P}(s'|s, a) \quad \text{for all } s' \in \mathcal{S}.$$

Now all that remains to be proven is the uniqueness aspect of this form. To that end, we first write the above set of equations in a matrix form

$$W = (1 - \gamma)D_0 + \gamma \mathcal{P}_{\pi_\theta}^T W, \tag{19}$$

where $W = (w(s_1), w(s_2), ....., w(s_{|\mathcal{S}|})) \in \mathbb{R}^{|\mathcal{S}|}$, $\mathcal{P}_{\pi_\theta}, D_{\pi_\theta} \in \mathbb{R}^{|\mathcal{S}| \times |\mathcal{S}|}$ where $(\mathcal{P}_{\pi_\theta})_{(s,s')} = \mathbb{P}^{\pi_\theta}(s'|s)$ and $D_{\pi_\theta}$ is a diagonal matrix whose every element correspond to $d_0$. It is easy to see from this form here that the row sum of this matrix is 1. We can simply re-write the above equation as follows,

$$(I_{|\mathcal{S}|} - \gamma \mathcal{P}_{\pi_\theta}^T)W = (1 - \gamma)D_0$$

$I_{|\mathcal{S}|}$ is the identity matrix of size $|\mathcal{S}|$. It now remains to prove that $(I_{|\mathcal{S}|} - \gamma \mathcal{P}_{\pi_\theta}^T)$ is invertible. We propose the following Lemma which proves the same result.

**Lemma 5.** *For $\gamma < 1$, The matrix $(I_{|\mathcal{S}|} - \gamma \mathcal{P}_{\pi_\theta}^T)$ is invertible.*

*Proof.* It is generally easier to prove that the transpose of this matrix is invertible. Consider $x \in \mathbb{R}^{|\mathcal{S}|}$ is a non-zero vector. We will now prove that $(I_{|\mathcal{S}|} - \gamma \mathcal{P}_{\pi_\theta})x$ is non-zero. To see that observe the infinty norm as follows,

$$\|(I_{|\mathcal{S}|} - \gamma \mathcal{P}_{\pi_\theta})x\|_\infty \geq \|x\|_\infty - \gamma \|\mathcal{P}^{\pi_\theta} x\|_\infty$$
$$\geq (1 - \gamma)\|x\|_\infty > 0$$

We get the first equation from the triangular inequality. We get the second equation from the fact that the row sum of the matrix $\mathcal{P}_{\pi_\theta}$ is 1. Thus, $\|\mathcal{P}_{\pi_\theta} \cdot x\|_\infty \leq \|x\|_\infty$. This implies that $(I_{|\mathcal{S}|} - \gamma \mathcal{P}_{\pi_\theta}^T)$ is invertible. This completes the proof. □

Since the matrix $(I_{|\mathcal{S}|} - \gamma \mathcal{P}_{\pi_\theta}^T)$ is invertible the solution is equal to $d_\gamma^{\pi_\theta}$. The solution also satisfies the constraint $\sum_s w(s) = \sum_s d^{\pi_\theta}(s) = 1$. This completes the proof for the first case.

**Case 2.** $\gamma = 1$
For $\gamma = 1$ we need to solve

$$(I - \mathcal{P}_{\pi_\theta}^T)W = 0$$

The proof now follows from (Zhang et al., 2020b, Theorem 1). Here we invoke Perron-Frobenius theorem which says the dimension of the left eigenspace of $\mathcal{P}_{\pi_\theta}$ corresponding to eigenvalue 1 is one-dimensional. Since $d_1^{\pi_\theta}$ belongs in that set, the solution of these set of equations as $W = \alpha d_1^{\pi_\theta}$. It is worth noting that only $\alpha = 1$ satisfies the second constraint which is $\sum_s w(s) = 1$. This completes the proof. □

### 9.5 Proof of Lemma 3

This proof is very similar to proof of Lemma 2.

*Proof.* Let us restate the optimization problem in Lemma 3 (equation 11):

$$\min_{w:\mathcal{S}\times\mathcal{A}\to\mathbb{R}^n} \left\{ \mathbb{E}_{(s',a')\sim d_\gamma^{\pi_\theta}} \|\nu(s',a')\|^2 + \frac{\lambda}{2}\left\| \mathbb{E}_{(s,a)\sim d_\gamma^{\pi_\theta}} [w(s',a')]\right\|^2 \right\}$$
$$\nu(s',a') := d_\gamma^{\pi_\theta}(s',a')(w(s',a') - \nabla_\theta \log \pi_\theta(a'|s')) - \gamma \sum_{s,a} d_\gamma^{\pi_\theta}(s,a)\mathcal{P}(s'|s,a)\pi_\theta(a'|s')w(s,a)$$

We note that it has two quadratic terms. As in Lemma 2, this loss function can only go to zero if and only if both the quadratic terms turn out to be zero. This implies that

$$\nu(s',a') = 0 \text{ for all } (s',a') \in \mathcal{S} \times \mathcal{A} \quad \text{and} \quad \mathbb{E}_{(s',a')\sim d_\gamma^{\pi_\theta}}[w(s',a')] = 0.$$

We take two cases.

**Case 1.** $\gamma < 1$

Similar to Lemma 2, for a finite state-action space we can re-write equation 11 in a linear form

$$(I_{|\mathcal{S}|\times|\mathcal{A}|} - \gamma \mathcal{P}_{\pi_\theta}^T) D_{\pi_\theta} W = D_{\pi_\theta} G \tag{20}$$

where, $W \in \mathbb{R}^{|\mathcal{S}| \cdot |\mathcal{A}| \times n}$ is the matrix with every row corresponding to $w(s, a)$ for each state-action pair $(s, a)$. Similarly, $G \in \mathbb{R}^{|\mathcal{S}| \cdot |\mathcal{A}| \times n}$ has its rows as $\nabla_\theta \log \pi_\theta$ for each state-action pair. Let $\mathcal{P}_{\pi_\theta}, D_{\pi_\theta} \in \mathbb{R}^{|\mathcal{S}| \cdot |\mathcal{A}| \times |\mathcal{S}| \cdot |\mathcal{A}|}$ where $(\mathcal{P}_{\pi_\theta})_{((s,a),(s',a'))} = \mathbb{P}^{\pi_\theta}(s', a'|s, a)$ and $D_{\pi_\theta}$ is a diagonal matrix whose every element correspond to $d_\gamma^{\pi_\theta}$ for each state-action pair. The ergodicity assumption implies that $D_{\pi_\theta}$ is invertible. Additionally, we also proved in Lemma 5 in Section 9.4 that $(I - \gamma \mathcal{P}_{\pi_\theta}^T)$ is invertible. This ensures that the solution of equation 20 is unique and equal to the log density gradient. Since, log density gradient always satisfies the constraint $\mathbb{E}_{(s,a) \sim d_\gamma^{\pi_\theta}}[\nabla_\theta \log d^{\pi_\theta}(s, a)] = 0$, the second constraint becomes redundant. This completes the proof.

**Case 2.** $\gamma = 1$

For $\gamma = 1$ we have the following set of equations

$$(I_{|\mathcal{S}| \times |\mathcal{A}|} - \mathcal{P}_{\pi_\theta}^T) D_{\pi_\theta} W = D_{\pi_\theta} G \quad \text{and} \quad e^T D_{\pi_\theta} W = 0.$$

Let us look at the equations column by column. The matrix $(I_{|\mathcal{S}| \times |\mathcal{A}|} - \mathcal{P}_{\pi_\theta}^T)$ is not invertible since the vector $d_1^{\pi_\theta}$ is in its nullspace. Since $\nabla_\theta \log d_1^{\pi_\theta}$ satisfies the first equation, every column can be written as $\nabla_\theta \log d_\gamma^{\pi_\theta} + v$ where $v$ is any vector in the span of $\{d_1^{\pi_\theta}\}$. This implies that $D_{\pi_\theta} v = \mathcal{P}_{\pi_\theta}^T D_{\pi_\theta} v$. invoking Perron-Frobenius theorem as in Lemma 9.4, we see that $v$ lies in the span of $\{d_1^{\pi_\theta}\}$. We complete the proof by trying to satisfy the second constraint $e^T D_{\pi_\theta} v = 0$ which gives us $v = 0$. This completes the proof. $\qquad \square$

### 9.6 Proof of Lemma 4

*Proof.* We will prove contraction first for $\gamma \in [0, 1)$.

**Proof of Contraction:** To prove contraction, we will show that given any arbitrary functions $U, V : \mathcal{S} \times \mathcal{A} \to \mathbb{R}^n$, their difference under the L1 norm of the distribution $d_\gamma^{\pi_\theta}$ is a contraction.

$$\sum_{s',a'} d_\gamma^{\pi_\theta}(s', a') |Y_\gamma \cdot U(s', a') - Y_\gamma \cdot V(s', a')|$$

$$= \sum_{s',a'} d_\gamma^{\pi_\theta}(s', a') |\gamma \sum_{s,a} d_\gamma^{\pi_\theta}(s, a)(U(s, a) - V(s, a))\mathcal{P}(s'|s, a)\pi_\theta(a'|s')|$$

$$\leq \gamma \sum_{s',a'} d_\gamma^{\pi_\theta}(s', a') \sum_{s,a} d_\gamma^{\pi_\theta}(s, a)|U(s, a) - V(s, a)|\mathcal{P}(s'|s, a)\pi_\theta(a'|s')$$

$$\leq \gamma \sum_{s,a} d_\gamma^{\pi_\theta}(s, a) \sum_{s',a'} d_\gamma^{\pi_\theta}(s', a') \, |U(s, a) - V(s, a)|\mathcal{P}(s'|s, a)\pi_\theta(a'|s')$$

$$\leq \gamma \sum_{s,a} d_\gamma^{\pi_\theta}(s, a) \sum_{s',a'} |U(s, a) - V(s, a)|\mathcal{P}(s'|s, a)\pi_\theta(a'|s')$$

$$\leq \gamma \sum_{s,a} d_\gamma^{\pi_\theta}(s, a)|U(s, a) - V(s, a)| = \gamma \|U - V\|_1^{d_\gamma^{\pi_\theta}}.$$

This completes the proof. The contraction property is useful in proving uniqueness and convergence which we prove next.

**Case 1.** $\gamma < 1$

Let $Y_\gamma^k$ denote that $Y_\gamma$ has been composed with itself $k$ times. We are interested in what happens when $\lim_{k \to \infty} Y_\gamma^k$.

$$Y_\gamma^k C_0 = \gamma^k D_{\pi_\theta}^{-1} (\mathcal{P}_{\pi_\theta}^T)^k D_{\pi_\theta} C_0 + \sum_{t=0}^{k-1} \gamma^t D_{\pi_\theta}^{-1} (\mathcal{P}_{\pi_\theta}^T)^t D_{\pi_\theta} G$$

Noting that $\mathcal{P}_{\pi_\theta}$ is a transition probability matrix so every element is bounded above by 1, and since $\gamma < 1$, we get that $\lim_{k \to \infty} \gamma^k D_{\pi_\theta} (\mathcal{P}_{\pi_\theta}^T)^k D_{\pi_\theta} C_0 = 0$. Focusing on the other terms as $k \to \infty$ we get the following,

$$\lim_{k \to \infty} \sum_{t=0}^{k-1} \gamma^t D_{\pi_\theta}^{-1} (\mathcal{P}_{\pi_\theta}^T)^t D_{\pi_\theta} G = D_{\pi_\theta}^{-1} (I - \gamma \mathcal{P}_{\pi_\theta}^T)^{-1} D_{\pi_\theta} G$$

From equation 20 we see that $\lim_{k\to\infty} Y_\gamma^k C_0 = \nabla_\theta \log d_\gamma^{\pi_\theta}$ which completes the proof. $\qquad\square$

## 9.7   Additional Details Linear-TD

For large problem spaces, it is difficult to learn $w(s', a')$ for all the state-action spaces. In that case, we use linear function approximation, see for example, Gelada & Bellemare (2019); Hallak & Mannor (2017). We choose a feature map $\Phi : \mathcal{S} \times \mathcal{A} \to \mathbb{R}^d$ and $\zeta \in \mathbb{R}^{d \times n}$ are the linear parameters that we wish to learn. We want to approximate $w(s, a)$ as $\zeta^T \Phi(s, a)$ for each state action pair $(s, a)$. The TD(0) algorithm with linear function approximation has the following update

$$\zeta_{k+1}^T \leftarrow \zeta_k^T + \alpha_k(\gamma \zeta_k^T \Phi(s_k, a_k) + g(s_k', a_k') - \zeta_k^T \Phi(s_k', a_k'))\Phi(s_k', a_k')^T, \qquad (21)$$

where $(s_k, a_k) \sim d_\gamma^{\pi_\theta}, s_k' \sim \mathcal{P}(\cdot|s_k, a_k), a_k' \sim \pi_\theta(\cdot|s_k')$ and $\alpha_k$ is the learning rate. To prove convergence, re-write equation 21 in the following linear form,

$$\zeta_{k+1} = \zeta_k + \alpha_k(A_{k+1}\zeta_k + g_{k+1})$$

where, $A_k := \gamma\Phi(s_k', a_k')(\Phi(s_k, a_k)^T - \Phi(s_k', a_k')^T), g_k := \Phi(s_k', a_k')(\nabla_\theta \log \pi_\theta(a_k'|s_k'))^T$. We further define,

$$A := \gamma\mathbb{E}[\Phi(s', a')(\Phi(s, a)^T - \Phi(s', a')^T) \,|\, (s, a) \sim d_\gamma^{\pi_\theta}, s' \sim \mathcal{P}(\cdot|s, a), a' \sim \pi_\theta(\cdot|s')],$$
$$g := \mathbb{E}[\Phi(s', a')(\nabla_\theta \log \pi_\theta(a'|s'))^T \,|\, (s, a) \sim d_\gamma^{\pi_\theta}, s' \sim \mathcal{P}(\cdot|s, a), a' \sim \pi_\theta(\cdot|s')]$$

To prove the convergence of this linear TD update rule, we make a few standard assumptions.

**Assumption 2.**     *1. The matrix $\Psi$ has linearly independent rows.*

*2. The matrix $A$ is non-singular.*

*3. The feature matrix $\Phi$ has uniformly bounded second moments.*

**Theorem 3.** *Under Assumption 2 and the fact that the learning rate satisfies the Robbins Monroe condition Robbins & Monro (1951) then the TD update equation 21 converges in probability to the solution*

$$\lim_{k\to\infty} \zeta_k = -A^{-1}b$$

*Proof.* The proof follows similar to (Zhang et al., 2020b, Theorem 2) which invokes (Borkar & Meyn, 2000, Theorem 2.2). We first re-write the updated equation in the following form

$$\zeta_{k+1} = \zeta_k + \alpha_k(A\zeta_k + g + (A_{k+1} - A)\zeta_k + (g_{k+1} - g)).$$

The proof follows almost equivalently from (Zhang et al., 2020b, Theorem 2) since the assumptions are the same. The above equation has separated the update into the deterministic part, $h(\zeta_k) := A\zeta_k + g$ and the Martingale part, $M_{k+1} := (A_k - A)\zeta_k + (g_k - g)$. To apply this theorem, we now need to show that the function $h(\zeta)$ is asymptomatically stable. For stability, we will now show that the matrix $A$ has all negative Eigenvalues. Consider any unit eigenvector $x$. We will now try to evaluate $x^T A x$.

$$x^T A x = x^T \mathbb{E}_{(s,a)\sim d_\gamma^{\pi_\theta}}[-\Phi(s, a)\Phi(s, a)^T + \gamma\Phi(s, a)\mathbb{E}_{s'\sim\mathcal{P}(\cdot|s,a),a'\sim\pi_\theta(\cdot|s')}[\Phi(s', a')]] x$$
$$= -\mathbb{E}_{(s,a)\sim d_\gamma^{\pi_\theta}}[x^T \Phi(s, a)\Phi(s, a)^T x] + \gamma[x^T \Phi(s, a)\mathbb{E}_{s'\sim\mathcal{P}(\cdot|s,a),a'\sim\pi_\theta(\cdot|s')}[\Phi(s', a')]]$$

Using Cauchy-Shwartz inequality on the second term. We now get the following,

$$x^T A x \leq -(1 - \gamma)\mathbb{E}_{(s,a)\sim d_\gamma^{\pi_\theta}}[x^T \Phi(s, a)\Phi(s, a)^T x] \leq 0$$

Since $A$ is non-singular (Assumption 2), the eigenvalues cannot be zero. Thus, the eigenvalues for this problem are strictly negative. This guarantees asymptotic stability of $h(\zeta)$ and completes the proof. $\qquad\square$

### 9.8 Proof of Theorem 1

*Proof.* The proof follows similar to to (Zhang et al., 2020b, Theorem 2) which invokes (Borkar & Meyn, 2000, Theorem 2.2). We first re-write the updates in equation 18 in matrix form as $d_{t+1} = d_t + \varepsilon_t(G_{t+1}d_t + h_{t+1})$ with $d_t := [\alpha_t, \beta_t, \tau_t^T]$ and $G_{t+1}, h_{t+1}$ are as follows,

$$
G_{t+1} = \begin{bmatrix} 0 & -A_t & -\lambda\Phi_t \\ A_t & -C_t & 0 \\ \lambda\Phi_t^T & 0 & -\lambda \end{bmatrix}, \quad h_{t+1} = \begin{bmatrix} 0 \\ -B_t \\ 0 \end{bmatrix}
$$

where, $A_t = (\Phi_t\Phi_t^T - \gamma\Phi_t(\Phi_t')^T), B_t = \Phi g_t^T, C_t = \Phi_t\Phi_t^T$. We can calculate the expectation for each of these matrices as follows (the expectation is taken over all possible values of $(s, a)$ distributed as $d_\gamma^{\pi_\theta}$),

$$
G = \mathbb{E}_p[G_{t+1}] = \begin{bmatrix} 0 & -A & -\lambda D_{\pi_\theta}{}^T\Phi \\ A & C & 0 \\ \lambda\Phi^T D_{\pi_\theta} & 0 & -\lambda \end{bmatrix}, \quad h = \mathbb{E}_{(s,a)\sim d_\gamma^{\pi_\theta}}[h_{t+1}] = \begin{bmatrix} 0 \\ -B \\ 0 \end{bmatrix}
$$

Where,

$$
A := \Psi(I - \gamma P_{\pi_\theta})D_{\pi_\theta}\Psi^T, \quad B := \Psi D_{\pi_\theta}G^T, \quad C := \Psi D_{\pi_\theta}\Psi,
$$

$$
E_p[\cdot] := \mathbb{E}_{(s,a)\sim d_\gamma^{\pi_\theta}, s'\sim\mathcal{P}(\cdot|s,a), a'\sim\pi_\theta(\cdot|s')}[\cdot]
$$

where $D_{\pi_\theta}$ is a diagonal matrix with the diagonal being $d_\gamma^{\pi_\theta}$. We will now prove the convergence of the linear function approximation case. We first separate the deterministic term $h(d_t)$ and the stochastic term $M_{t+1}$

$$
d_{t+1} = d_t + \varepsilon_t(\underbrace{Gd_t + g}_{h(d_t)} + \underbrace{(G_{t+1} - G)d_t + (g_{t+1} - g)}_{M_{t+1}})
$$

From here, the proof follows equivalently to (Zhang et al., 2020b, Theorem 2), with exactly same assumptions. The remaining part of the proof requires us to show that the function $h(d)$ is symptomatically stable. We show that by demonstrating that the eigenvalues of the matrix $G$ are strictly negative. Let $v$ be an arbitrary eigenvector of $G$ corresponding to an arbitrary eigenvalue $\nu$, then

$$
\begin{aligned}
\mu &= v^T G v \\
&= v_2^T A v_1 + \lambda v_3^T \Phi^T D_{\pi_\theta} v_1 - v_1^T A v_2 - \lambda v_1^T \Phi^T D_{\pi_\theta} v_3 - v_2^T C v_2 - \lambda v_3^T v_3 \\
&= -\lambda v_3^T v_3 - v_2^T C v_2 \leq 0
\end{aligned}
$$

Now, we just need to show that the eigenvalue is not zero. We will prove that using contradiction. Assume that there exists a $v$ such that $Gv = 0$ and $v \neq 0$. This implies that,

$$
\begin{aligned}
&- A v_2 - \lambda\Phi^T D_{\pi_\theta} v_3 = 0 \\
&A v_1 = 0 \\
&\lambda D_{\pi_\theta}{}^T \Phi v_1 - \lambda v_3 = 0
\end{aligned}
$$

Since $A$ is a non-singular matrix (Assumption 1) we have $v_1 = 0$ which implies that $v_3 = 0$ from the third equation. This leaves us with $Av_2 = 0$ which leaves us with $v_2 = 0$, which is a contradiction. This finishes the proof. It now is proven that the eigenvalue is strictly negative. This completes the proof. $\square$

### 9.9 Proof of Theorem 2

*Proof.* The proof of this Theorem is almost similar to the proof of Zhang et al. (2020b) except the need to bound different terms. To simplify the proof, we first lump the maximization and minimization variables together and update these parameters in matrix form. We call the grouped maximization variables $y = (\beta, \tau)$ and minimization variable $\alpha$. We re-write Algorithm 1 in matrix form as follows (in Algorithm 3), where,

---

**Algorithm 3** Projected Log Density Gradient

---
1: **for** $i = 1, 2, ..., n$ do:
2: $\alpha_{t+1} = \Pi_X(\alpha_t - \varepsilon_t(G_{1,t}y_t))$
3: $y_{t+1} = \Pi_{Y,Z}(y_{t+1} + \varepsilon_t(G_{2,t}\alpha_t + G_{3.t}y_t + G_{4,t}))$
4: **Return** $\bar{\alpha}, \bar{y}$
    Where, $\bar{\alpha} = \dfrac{\sum_{i=1}^n \varepsilon_i \alpha_i}{\sum_{i=1}^n \varepsilon_i}$ , $\bar{y} = \dfrac{\sum_{i=1}^n \varepsilon_i y_i}{\sum_{i=1}^n \varepsilon_i}$

---

$$G_{1,t} := \begin{bmatrix} -(\Phi_t\Phi_t^T - \gamma\Phi_t\Phi_t'^T) & -\lambda\Phi_t \end{bmatrix}$$
$$G_{2,t} := \begin{bmatrix} \Phi_t\Phi_t^T - \gamma\Phi_t\Phi_t^{'T} \\ \lambda\Phi_t^T \end{bmatrix}, G_{3,t} := \begin{bmatrix} -\Phi_t\Phi_t^T & 0 \\ 0 & -\lambda \end{bmatrix}$$
$$G_{4,t} := \begin{bmatrix} -\Phi_t g_t^T \\ 0 \end{bmatrix}.$$

We also project our variables $\alpha_t, y_t$ on the closed and convex sets $X \subset \mathbb{R}^{d\times n}, Y \subset \mathbb{R}^{d\times n} \times \mathbb{R}^{1\times n}$. We output the weighted average $\bar{\alpha}$ and $\bar{y}$. Before, proposing a sample complexity bound, we first define the optimization gap $\epsilon_g(\alpha, y)$ as follows,

$$\epsilon_g(\alpha, y) := \max_{y' \in Y} L(\alpha, y') - \min_{\alpha' \in X} L(\alpha', y)$$

Note that $\epsilon_g(\alpha_*, y_*) = 0$, where $\alpha_*, y_*$ are the solutions to the min-max problem. From here, on the proof follow almost similarly to Zhang et al. (2020b). The proof follows from (Liu et al., Proposition 3) and (Zhang et al., 2020b, Proposition 2), both of which rely on (Nemirovski et al., 2009, Proposition 3.2) to state the $O\left(\sqrt{\frac{1}{n}}\right)$ bound. Proposition 3.2 in Nemirovski et al. (2009) says that set $X, Y, Z$ should be closed, convex and bounded sets, which is part of our assumption. Our min-max loss function is Lipschitz continuous and the minimization problem is convex and the maximization problem is concave. It can also be seen that both the primal and dual form of the optimization has equal optimal values $\alpha_*, y_*$. We now proceed to apply the bound proposed by Nemirovski et al. (2009). To that end, we first need to bound certain terms. We define $D_\alpha$ and $D_Y$ as follows,

$$D_\alpha = \max_{x \in X} \|x\|^2 - \min_{x \in X} \|x\|^2$$
$$D_Y = \max_{y \in Y} \|y\|^2 - \min_{y \in Y} \|Y\|^2$$

From Assumption 1 which says the second moment of all the features is bounded we can similarly write the following bound

$$\mathbb{E}[\|G_{i,t} - G_i\|^2] \le \sigma_i^2 \quad \forall i \in \{1, 2, 3, 4\}.$$

Therefore, we obtain bounds for the stochastic sub-gradient $G_\alpha(\alpha, y), G_Y(\alpha, y)$ as follows,

$$G_\alpha(\alpha, y) = G_{1,t}y_t$$
$$G_Y(\alpha, y) = (G_{2.t}\alpha_t + G_{3,t}y_t + G_{4,t})$$

and for their second moment as follows,

$$\mathbb{E}[\|G_\alpha(\alpha, y)\|^2] = \sigma_1^2 D_Y^2 + \sigma_1^2\|\bar{G}_{1,t}\|^2 \le C_\alpha$$
$$\mathbb{E}[\|G_Y(\alpha, y)\|^2] = \sigma_2^2 D_Y^2 + \sigma_2^2\|\bar{G}_{2,t}\|^2 + \sigma_3^2 D_Y^2 + \sigma_3^2\|\bar{G}_{3,t}\|^2 + \sigma_4^2\|\bar{G}_{4,t}\|^2 \le C_Y,$$

where we used the fact that $\mathbb{E}[\|x\|^2] \le \mathbb{E}[\|x - \mathbb{E}[x]\|^2] + \|\mathbb{E}[x]\|^2$. If we now follow the procedure as proposed by Nemirovski et al. (2009) we can define $M_*$ as follows,

$$M_*^2 = 2C_\alpha^2 D_{\alpha^2} + 2C_Y^2 D_Y^2$$

If we fix the learning rate as $\varepsilon_t = \frac{c}{M_* \sqrt{t}}$ for any positive constant $c$. We can now bound the optimality gap with probability at least $1 - \delta$ using (Nemirovski et al., 2009, Proposition 3.2)

$$\epsilon(\bar{\alpha}, \bar{y}) \leq \frac{5}{n}(8 + 2\log\frac{2}{\delta})M_* \max\{c, \frac{1}{c}\}$$

This completes the proof. □