# OpenReview forum: "Towards Provable Log Density Policy Gradient"
_TMLR — Accepted by TMLR_

### Review · Reviewer_LKDn · 2024-06-28

**Summary Of Contributions:**

The paper presents a novel technique for estimating the policy gradient in the case of an average reward scenario. Existing techniques rely on $\gamma < 1$ to obtain an algorithm, but as previous work has shown, this results in a biased gradient when using a $Q_\gamma$ function to estimate the average reward policy gradient.

The paper presents both a full estimate based on the backwards state transition probability and a more application friendly sample based version. Finally, the efficacy of the approach is verified in a small scale grid world experiment.

**Audience:**

Yes

**Claims And Evidence:**

Yes

**Requested Changes:**

Nice to have: In their introduction, the authors imply that most RL work is interested in the average reward formulation. I believe that this claim is slightly too strong and can lead to confusion: the policy gradient is (as far as I know) not biased if the goal is indeed to optimize for a discounted return (although almost no algorithm that is in practical use uses the discounted state visitation distribution, so I agree with the authors here that there is are come non-trivial oversights). The authors are correct that previous work has not been careful in differentiating between the goal stated in the respective works and those optimized by the presented algorithms, but I would argue that both optimizing for a discounted or average reward scenario is a valid goal. I would encourage the authors here to clarify the introduction to highlight under what conditions the policy gradients are biased.

Nice to have:
(slightly) larger experiment with actual function approximation

Very good to have:
The paper has quite a few typos and grammatical errors, another editorial pass would be good.

Critical:
Rework and expand the explanation of the method, the purpose of introduction for each step in the derivations, etc. I am currently rating the paper as having no "accurate, convincing and clear evidence" not because I have serious doubts about the correctness of the derivations, but because I believe that the writing can and should be seriously improved before acceptance. I want to stress that I believe that the contribution is both valuable and correct (as far as I can tell) and I commend the authors for their work but simply not really easily accessible in its current form.

**Strengths And Weaknesses:**

Strengths:
The authors present a thorough derivation of both the problems with existing approaches, and their solution. By providing both a population version, a sample based estimation approach and a sample complexity bound, they cover a good set of interesting questions relating to a novel approach.

As far as I can tell, all proofs are correct (although see the following).

Weaknesses:
The exposition and presentation of the paper makes following the core derivations and results hard. The proofs in the appendix do not all restate the statements which are being proven, and equations across the paper are referenced requiring a reader to jump back and forth a lot. Furthermore, derivation steps could be shown with more detail, reconstructing some of the steps has taken me a non-trivial amount of time. The authors also at several points refer to proofs in other papers. I would prefer if the authors either restated results from other papers they are using explicitly, or rewrite the proofs in full here. The paper should be readable by itself as much as possible.

One concrete example for unclear exposition is the introduction of Lemma 2: the role of $w(s)$ is not explicitly stated in the leadup to the lemma, nor after. This means the reader has to intuit by themselves what the purpose of its introduction is. Overall, I would encourage the authors to thoroughly rework the expository text for the mathematical results. I first assumed the role is simply to approximate $d^\pi$, but later the same variable is used to approximate the gradient of $d^\pi$, so it is simply used as the "variable to learn" in each lemma?

In the text after Lemma 2, the exposition jumps back to the term $\nabla d$ without any transition or even a paragraph break. This is related to the previous problem. From context I reconstructed that the authors intended to present to successive lemmas about the learnability of $d$ and $\nabla d$ respectively, but this was not stated clearly.

The experiments are very small scale, with tiny grid worlds. It would be interesting to see if the sample complexity shown by the authors is sufficient to scale up to slightly more complex problems, like larger gridworlds, standard testing benchmarks such as cliffwalk, chainwalk, garnets, or maybe mountain car.
In the same spirit: as the authors motivated their sample based approach by highlighting how it enables function approximation, I think it would be a good idea to show small results with function approximation, such as tiling features over gridworlds. While they claim that they use function approximation, they are using one-hot features per state-action pair, which is basically equivalent to a tabular method.

---

> ### Author Response · Authors · 2024-08-31
> **Response to LKDN**
>
> Dear Reviewer LKDn,
>
> Thank you for taking the time to understand and read the paper in so much detail. Your suggestions are very valuable. We would like to address the points you make here and will revise our draft based on your suggestions.
>
> **Average Reward Scenario**: Policy gradient theorem has been defined for all values of discounting factor. In principle, one can use it for updating policy for average reward scenarios. In many RL settings, we are interested in maximising for average reward scenarios. For example, look at the training curves in soft-actor critic paper and/or PPO paper [1]. The only caveat in modern reinforcement learning literature is that, the Bellman equation is not a contraction anymore for gamma = 1, so classical Q-function estimation techniques cannot be used. As a compromise, RL practioners typically learn a Q-function for discounting factor smaller than one and use that to estimate the gradient. This leads to an error in policy gradient estimation, correcting for it could not only lead to better convergence but also could lead to an increase in sample efficiency.
>
> **Paper Organization**: Thank you for your detailed comments on our paper writing and organization. We will incorporate and  address the key changes that you mentioned in our draft.
>
> **Experiments**: Our paper is focused more on proposing our log density gradient algorithm, which now allows us to exactly calculate policy gradients for average reward formulations accurately and provably. To that end, our experimental results aim to showcase proof-of-concept results that demonstrate log density gradient’s effectiveness. Therefore in our experimental results, we focus solely on linear function approximation cases. To that end, we use one-hot encoding of the state and action spaces as the features. Running the experiments for larger state-spaces (bigger gridworld) as well study of other domains you mentioned is the subject of follow up work. A common issue that we feel with our experiments is the need for linear features which may not be readily available.
>
>
> [1]: Schulman, J., Wolski, F., Dhariwal, P., Radford, A., and Klimov, O. Proximal policy optimization algorithms.

---

### Review · Reviewer_5zdo · 2024-07-02

**Summary Of Contributions:**

This paper uses the log density gradient idea to correct for the residual error in policy gradient estimation. And then a model-free TD based method is used to approximate the policy gradient with some theoretical guarantee. To overcome the practical inefficiency issue, this work further implements the min-max optimization idea that works under the on-policy samples and present some theoretical results under the linear function class assumption, which enables their method to work for the linear MDP problems. Empirical results are presented to validate the high efficiency of the proposed method.

**Audience:**

Yes

**Claims And Evidence:**

Yes

**Requested Changes:**

1. I feel that given the existing literature in Morimura et al. and Zhang et al., it may not be fair to state you "propose" the log density gradient and "propose" the min-max optimization, since your methods are adapted from the existing works.
2. It would be better to summarize the min-max optimization methods in the existing literature, e.g. Zhang et al.. Since your method also adapts this idea.
3. Including a recent baseline or adding some real experiments can make the experimental results stronger.
4. How difficult it is to extend from the average reward scenario to the general case over a range of discounting factor $\gamma \in [0,1]$? What are the major challenges in this extension? Please briefly explain this issue in your work.
5. A brief explanation on Assumption 1 is necessary.
6. TD($\lambda$) is not explained in the main paper (but in appendix) but TD(0) is used in the conclusion. Please briefly explain this notation before using it.

**Strengths And Weaknesses:**

Strengths:
1. This work has a comprehensive summary of the existing literature.
2. I am not an expert in this area and hence I didn't check the proof in Appendix in detail, but all the arguments look sound to me.

Weaknesses:
1. I feel the presentation of this work could be improved.
1.1 There are some typos, e.g.: in page one "we empirically demonstrate that this error 'in' indeed significant"; in page 5, there is no verb in the sentence "Thus, effectively demonstrating that TD-updates..."
1.2 Some parts are confusing and hard to understand: e.g. in page 5 sentence "Thus, effectively demonstrating that TD-updates...", I don't understand why your proof (converging to a unique solution) could showcase the drawback of the method from Morimura. Will this result indicate the drawback of your methods as well?
2. I feel the baseline in your experimental results is a little bit too weak. It would be better to compare your method with some modern policy gradient based algorithms but not just REINFORCE which is about 30 years old. And real-world datasets should also be used in your experiments.

---

> ### Author Response · Authors · 2024-08-31
> **Response to Reviewer**
>
> Dear Reviewer 5zdo,
>
> Thank you for taking the time to read and review our manuscript. Your feedback is very valuable and we will revise our draft accordingly. The responses to your concerns are as follows, listed in the same order.
>
> - We agree with your point that the experimental results could be tested on a larger set of environments and across an extensive range of baselines. The key contribution of our log density gradient is to propose another approach to exactly calculate policy gradient for average reward formulation in reinforcement learning. This is something that the reinforcement learning community currently approximates using classical policy gradient formulation. To that end, we believe the best way to perform an apple-to-apple comparison would be by comparing the basic form of log density gradient algorithm with the basic form of classical policy gradient theorem. This is what has been shown and thoroughly evaluated in our experimental results.
>
> - All our results hold for all $\gamma \in [0,1]$. We have made this fact more clear in the revised draft.

---

### Review · Reviewer_MfXw · 2024-08-13

**Summary Of Contributions:**

The paper re-introduces a the method called "log density gradient" in Morimura 2010 for estimating policy gradients in reinforcement learning, based on the gradient of log density. The authors show that the Morimura's method corrects for a residual error term that is often ignored in classical policy gradient implementations. The author derived the log density gradient in discounting setting, and a new optimization method to estimate the gradient of log density without sampling from backward Markov chain. Theoretical analysis is provided, proving convergence and uniqueness properties of the min-max optimization, as well as sample complexity bounds. These are novel contribution and is not included in the Morimura paper.

**Audience:**

Yes

**Claims And Evidence:**

Yes

**Requested Changes:**

See the three request in the weakness 1-3.

**Strengths And Weaknesses:**

Strengths:

This paper presents a theoretically grounded approach to policy gradient estimation in reinforcement learning. This paper further consolidate the theoretical foundation of the Morimura's method (log density gradient method). The log density gradient is so different from classical policy gradient and does not suffer from the residual error in classical policy gradient that it worth more attention of the community.

The min-max optimization formulation is particularly noteworthy as it overcomes the limitation of requiring backward Markov chain samples, making the method more practical for real-world applications.

Weakness:

1. This writing of mathematical equations and theorem statement should be improved for its rigorousness. For examples: In the definition of operator $Y_\gamma$ between Eq 13 and 14, $(s, a)$ should be sampled from the backward MC. The textual part of Proposition 1 and Proposition 2 should be written in a more formal language, and it should be self-contained: i.e. no "mentioned in Eq. x".

2. This paper should have a complete policy updating algorithm psedu-code or description. In the current version, after Section 3, the whole problems seems becoming how to estimate log density gradient. After solving that, the paper need to go back discuss what's the policy gradient and update rule, what's the effect of using an approximation of the log density gradient in the policy gradient.

3. The paper should discuss more limitation of the log density gradient methods in comparison with classical PG. For example, the TD method of log density gradient updates a d-dimensional vector function, where d is the number of policy parameter. Standard TD (to estimate the Q(s,a) in PG theorem) only need to update a real-valued function. For modern NN, d is huge. This is a significant concern of the scalability of the method.

4. The experimental domain is a 5x5 grid world.

---

> ### Author Response · Authors · 2024-08-31
> **Response to Reviewer MfXw**
>
> Dear Reviewer MfXw,
>
> Thank you for taking the time to read and review our manuscript. Your feedback is very valuable and we will revise our draft accordingly. We look forward to hearing more of your feedback in the revised draft.
>
>  Paper Organization:
>
> **Improving writing**: Thank you for pointing these out, we have made the changes to the revised draft.
>
> **Pseudo code**: The pseudo code for the algorithm is included in Algorithm 2 on page 11. It has been added to the draft.
>
> **Limitations**: We have a limitations section in the paper, which discusses the limitations at length
> Experimental results: Although experimental results on larger set of environments are warranted. Our paper is focused more on proposing our log density gradient algorithm, which now allows us to exactly calculate policy gradients for average reward formulations accurately and provably. To that end, our experimental results aim to showcase proof-of-concept results that demonstrate log density gradient’s effectiveness. More experiments which focus on scalability of log density gradient requires implementing variance reduction techniques to improve performance and is the subject of follow up work.

---

### Decision · Action_Editor_x5oG · 2024-10-15

**Recommendation:** Accept as is

**Comment:**

The authors already improved the organization following the reviewers suggestions in the revised version.

Please change the color of the revised component to normal.

**Audience:**

Yes. The reinforcement learning community will be interested in this work.

**Claims And Evidence:**

This paper reveal the recursion structure on the log density gradient, based on which a policy gradient estimation algorithm has been proposed. All the reviewers agreed that the idea is novelty and interesting.

The paper can be further improved if comprehensive empirical study can be provided.